# Chitosan-Based Edible Films as Innovative Preservation Tools for Fermented and Dairy Products

Fadime Seyrekoğlu [1],[*] and Esra Efdal [2]

1    Department of Food Processing, Suluova Vocational School, Amasya University, Suluova 05500, Amasya, Türkiye

2    Department of Biotechnology, Instute of Science, Amasya University, Suluova 05500, Amasya, Türkiye; esraefdal@gmail.com

*    Correspondence: fadime.tokatli@amasya.edu.tr

**Abstract**

Extending the shelf life and ensuring microbial stability of processed foods are key objectives in the food industry. In this study, edible films containing chitosan, chitosan + thyme (*Thymus vulgaris*) oil, and chitosan + rosemary (*Rosmarinus officinalis*) oil were applied to traditional and industrial Cecil cheese using the dipping method, with control groups for each production type. Samples were stored at $4 \pm 1$ °C for 45 days, and physical (color, water activity, and texture), chemical (pH, acidity, and dry matter), microbiological (total aerobic mesophilic bacteria, yeast-mold, coliforms, and lactic acid bacteria), and sensory analyses were performed on days 1, 15, 30, and 45. Results indicated that chitosan-based films effectively limited microbial growth, with the chitosan + rosemary oil combination being particularly effective in reducing microbial load and maintaining textural stability. Traditional cheeses achieved higher overall acceptability, while purchase intent was greater for industrial products. Coated samples exhibited slower pH decline and more stable dry matter content; industrial cheeses retained moisture more effectively. Texture profile analysis showed more stable chewiness and springiness values in coated samples. In conclusion, natural edible films represent an effective approach for extending shelf life and preserving quality, particularly in traditional cheeses with fibrous structures and shorter shelf lives.

**Keywords:** chitosan; essential oils; Cecil cheese; microbiological quality; sensory evaluation; bioactive coating

## 1. Introduction

Cheese is a nutrient-rich dairy product with substantial cultural and economic value, leading to the production of numerous varieties worldwide, including in Türkiye [1]. Traditional cheeses, such as Cecil cheese, are emblematic of regional heritage, possessing distinctive aroma, flavor, and texture profiles [2]. Originally unique to Eastern Anatolia, Cecil cheese is now produced across the country [3]. Its structure, characterized by its fibrous, stringy texture obtained through kneading and stretching [4], is influenced by lactic acid fermentation and scalding temperature [5]. However, its high moisture (55–60%) and low salt content, combined with traditional production methods, make it highly perishable and susceptible to pathogenic microorganisms such as Listeria monocytogenes and Staphylococcus aureus, as well as yeasts and molds [6,7]. These factors not only shorten shelf life but also raise food safety concerns and cause economic losses [8].

Growing consumer demand for natural, minimally processed foods and the environmental drawbacks of synthetic packaging have accelerated the search for sustainable

preservation methods [9]. Edible films and coatings, applied as thin biodegradable layers, have emerged as promising solutions [10]. They act as selective barriers, reducing moisture loss, controlling gas exchange, and slowing oxidative reactions, while preserving sensory quality [11]. The properties of these films depend on their composition: polysaccharides (e.g., chitosan and alginate) offer gas barrier capabilities, proteins (e.g., whey protein) provide mechanical strength, and lipids (e.g., waxes) enhance moisture resistance [12,13]. In recent years, the use of these biopolymers—alone or in combination—has expanded, with growing applications of edible films and coatings in cheese preservation [14]. Among these, chitosan stands out for its antimicrobial activity derived from its cationic structure, making it suitable for short-shelf-life cheeses [15].

The concept of active packaging—enhancing films with bioactive agents—has further improved preservation efficiency [16]. Essential oils from thyme (*Thymus vulgaris*) and rosemary (*Rosmarinus officinalis*), rich in antimicrobial and antioxidant phenolics such as carvacrol, thymol, and 1,8-cineole, have proven effective in controlling microbial growth and delaying lipid oxidation in cheese [17–20]. While edible coatings have been applied to cheeses like Feta, Kashar, and Ricotta [21,22], research on fibrous cheeses, particularly Cecil cheese, remains scarce. Its filamentous structure increases surface exposure, heightening susceptibility to spoilage. This study aimed to develop chitosan-based edible coatings enriched with thyme and rosemary essential oils, evaluate their antimicrobial activities, and determine their effects on the physicochemical, microbiological, textural, and sensory qualities of Cecil cheese produced by traditional and industrial methods. The findings are expected to identify the most effective coating for each production method and contribute to safe, natural, and sustainable preservation strategies for traditional cheeses.

## 2. Materials and Methods

### 2.1. Materials

In this study, raw milk used for the traditional production of Cecil cheese and industrially produced Cecil cheese was both obtained from Dalgıçlar Farm (EREN Gıda San. ve Tic. Ltd. Şti. Çorum, Türkiye). The rennet used in traditional cheese production (Evde Şirden Peynir Mayası) was supplied by MAYSA Gıda San. ve Tic. A.Ş. Homemade yogurt and lemon juice were used during traditional production to initiate fermentation and adjust acidity. The starter cultures used in cheese production were obtained in powdered form as commercial preparations of the "Danisco" brand, supplied by Türker Endüstri Teknik Makine ve Tic. Ltd. Şti. The cultures were prepared in accordance with the procedures indicated on the manufacturer's packaging and were applied as mixed cultures during cheese production. The culture compositions employed in this study were as follows: Culture 1: *Lactococcus lactis* subsp. *lactis* (ATCC 19435) + *Lactococcus lactis* subsp. *cremoris* (**ATCC 19257**) + *Lactococcus lactis* subsp. *diacetylactis* (ATCC 13675), Culture 2: *Lactococcus lactis* subsp. *lactis* (ATCC 19435) + *Lactococcus lactis* subsp. *cremoris* (ATCC 19257), and Culture 3: *Streptococcus salivarius* subsp. *thermophilus* (ATCC 19258).

For the preparation of edible films, chia seeds were purchased in bulk from a local herbal store, while chitosan (degree of deacetylation of 85%) was obtained from TİENS İç ve Dış Tic. Ltd. Şti. Thyme and rosemary essential oils used in film formulations were acquired from the brand KIRINTI (Kırıntı Baharat Hay. Tar. Kozm. Gıda Tem. MLz. İnş. Elek. Tur. San. Tic. Ltd. Şti.). Vegetable glycerin, used as a plasticizer, was of the brand VANCE and supplied by Mutlukal Gıda San. ve Tic. A.Ş. Sodium hydroxide (NaOH) and acetic acid were used for pH adjustments.

## 2.2. Methods

### 2.2.1. Edible Film Preparation

To determine the most suitable edible films for the coating of Cecil cheese, eight different groups of edible film formulations were initially prepared. These combinations are presented in Table 1.

**Table 1.** Edible film groups.

| Group Code | Material Composition |
| --- | --- |
| A | Chia seed |
| C | Chitosan |
| A1 | Chia seed + Rosemary oil |
| C1 | Chitosan + Rosemary oil |
| A2 | Chia seed + Thyme oil |
| C2 | Chitosan + Thyme oil |
| AC1 | Chia seed + Chitosan + Rosemary oil |
| AC2 | Chia seed + Chitosan + Thyme oil |

Edible film preparation with chia seeds.

During the production of edible film from chia seeds, the method proposed by Çelik (2020) was modified and applied [23]. A total of 30 g of chia seeds was weighed into a beaker, and distilled water was added at a 1:20 ratio. The pH of the mixture was adjusted to 8 using 0.2 M NaOH solution. The hydration process was carried out at 80 °C for 2 h using a magnetic stirrer. At the end of the process, the solution was poured evenly onto a drying tray and dried at 65 °C in an oven for 20 h. After drying, the chia seed residue was separated using a fine sieve. The dried chia seed mucilage that passed through the sieve was weighed. To form the film, 500 mL of distilled water (1% $w/v$) was added to 0.5 g of the obtained dry mucilage. The mixture was mechanically stirred at 25 °C for 2 h. Subsequently, the pH of the solution was adjusted to 9 using 0.1 M NaOH. Glycerol (vegetable-based), at 1% of the total solution volume (5 mL), was added as a plasticizer. The solution was then stirred at 80 °C for 30 min using a magnetic stirrer.

For five different combinations, 500 mL of the solution was divided into five portions of 100 mL each. For the pure chia seed formulation, 10 mL of the 100 mL portion was reserved for microbiological analysis, and the remaining solution was evenly poured into four petri dishes. The films were allowed to dry in an oven at 35 °C for 3–4 days.

Combination of Chia Seed and Essential Oils

For the chia seed and rosemary oil combination, 100 mL of the chia seed solution was taken, and 2% (200 μL) rosemary essential oil was added. The prepared mixture was homogenized using a magnetic stirrer at 40 °C for 15 min. After homogenization, 10 mL of the film-forming solution was reserved for microbiological analysis. The remaining solution was evenly poured into four petri dishes for film formation. The petri dishes were placed in an incubator at 35 °C and allowed to dry for 3–4 days.

For the chia seed and thyme oil combination, 100 mL of the chia seed solution was taken, and 2% (200 μL) thyme essential oil was added. The mixture was then homogenized using a magnetic stirrer at 40 °C for 15 min. Following homogenization, 10 mL of the film solution was set aside for microbiological analysis. The rest of the solution was evenly distributed into four petri dishes, which were then placed in an incubator at 35 °C to dry for 3–4 days.

Production of Edible Films with Chitosan

The edible film production process was based on the method proposed by Akat (2023) [24]. In this context, 10 g of chitosan was weighed and gradually dissolved in 1% acetic acid solution. The dissolution process was carried out in a controlled manner using a magnetic stirrer. Vegetable-based glycerol (0.06 mL/100 mL) was added to the chitosan solution as a plasticizer. The solution was then stirred on a magnetic stirrer at room temperature for one hour to achieve homogeneity. Following the mixing process, 10 mL of the prepared film solution was taken for microbiological analysis, while the remaining solution was poured equally into four petri dishes. The petri dishes were dried in an incubator at 35 °C for 3–4 days for film formation (Figure 1).

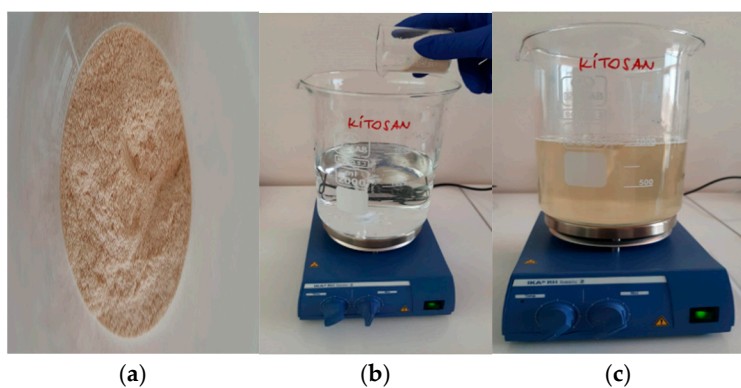

|        |        |        |
| :----: | :----: | :----: |
| (**a**) | (**b**) | (**c**) |

**Figure 1.** Preparation of chitosan film solution. (**a**) Chitosan (**b**) Addition of chitosan to pure water (**c**) Homogenization of chitosan and water.

Combination of Chitosan and Essential Oils (Rosemary Oil, Thyme Oil)

Chitosan and Rosemary Oil Combination: From the prepared chitosan solution, 100 mL was separated, and 2% (200 μL) rosemary oil was added. The mixture was stirred at 40 °C on a magnetic stirrer for 15 min. After the process, 10 mL was reserved for microbiological analysis, and the remaining solution was poured equally into four petri dishes. The petri dishes were dried in an incubator at 35 °C for 3–4 days.

Chitosan and Thyme Oil Combination: Similarly, 100 mL of the chitosan solution was taken, and 2% (200 μL) thyme oil was added. The mixture was stirred at 40 °C on a magnetic stirrer for 15 min. Afterward, 10 mL was separated for the microbiological analysis, and the remaining solution was equally distributed into four petri dishes and dried at 35 °C in an incubator for 3–4 days.

Combination of Chitosan, Chia Seed, and Essential Oils (Rosemary Oil and Thyme Oil)

Chitosan, Chia Seed, and Rosemary Oil Combination: 50 mL of the prepared chitosan solution and 50 mL of chia seed solution were mixed. Then, 2% rosemary oil was added and stirred at 40 °C for 15 min using a magnetic stirrer. After the process, 10 mL of the 100 mL solution was taken for microbiological analysis. The remaining solution was poured equally into four petri dishes and dried at 35 °C for 3–4 days.

Chitosan, Chia Seed, and Thyme Oil Combination: Similarly, 50 mL of chitosan solution and 50 mL of chia seed solution were combined, and 2% thyme oil (200 μL based on 10 mL total) was added. The mixture was stirred for 15 min at 40 °C using a magnetic stirrer. Afterward, 10 mL was separated for microbiological analysis. The remaining solution was distributed equally into four petri dishes and dried at 35 °C in an incubator for 3–4 days.

When compared to the PET/PANI nanocomposite production method reported in the literature [25], the chitosan-based coating method employed in this study shares com-

mon steps such as solution preparation and controlled drying but differs in composition, target application area, and evaluation parameters. While the PET/PANI method uses in situ chemical oxidative polymerization to produce conductive coatings for electronic applications, the present study develops a low-temperature, edible film with antimicrobial properties suitable for direct application to food surfaces.

### 2.2.2. Production of Cecil Cheese

#### Traditional Production

Cecil cheese production in Türkiye varies by region in terms of raw material selection, processing techniques, and ripening methods. While the fundamental characteristics of cheeses produced in Erzurum, Kars, Ardahan, and Ağrı are similar, local practices differ. For instance, cow milk is commonly used in Erzurum and Ardahan, whereas mixtures of sheep and goat milk are preferred in Kars and Ağrı depending on season al availability [23,26]. In Erzurum, the scalding process is typically conducted at higher temperatures for longer durations, while in Kars and Ardahan, shorter scalding times and smaller curd sizes are used [27,28]. Salting methods also vary: in Erzurum and Ağrı, cheeses are dry-salted and dried, whereas in Kars and Ardahan, they are stored in brine for extended periods [29]. These variations affect the cheese's moisture, salt content, and microbial stability. In this study, the traditional method commonly practiced in Erzurum was adopted (Figure 2).

#### Industrial Production

Industrial-scale Cecil cheese production aims to ensure hygiene, product standardization, and extended shelf life. Processing is carried out in closed-circuit systems from pasteurization to packaging, with continuous monitoring of critical control points such as temperature, pH, and salt concentration. Standardized milk is coagulated with starter culture and rennet, and the resulting curd is scalded under controlled conditions to develop the fibrous structure. The cheese is brined at defined concentrations and durations, then dried and packaged under vacuum or modified atmosphere. This method preserves the traditional fibrous texture while ensuring microbiological safety and a prolonged shelf life (Figure 3).

Detailed information regarding traditional and industrial production is presented in the Supplementary Materials section.

#### Preparation of Edible Film Solutions for Coating Cecil Cheese

The preparation of edible films was based on the procedure proposed by Akat (2023) for chitosan solution preparation [24]. Accordingly, 10 g of chitosan was weighed and dissolved in a 1% acetic acid solution to prepare the chitosan solution. The dissolution was carried out by slowly adding chitosan to the solution under magnetic stirring until fully dissolved. Plant-based glycerol was added as a plasticizer to the obtained chitosan solution at a ratio of 0.06 mL per 100 mL of solution. After the addition of glycerol, the mixture was stirred at room temperature on a magnetic stirrer for one hour until a homogeneous structure was achieved. This prepared film base solution was used in the edible film coating applications (Figure 1).

For the preparation of the chitosan and rosemary oil combination, the previously prepared chitosan solution was used as a base. Rosemary oil was added to the solution at 2%. To ensure uniform dispersion of the oil in the solution, the mixture was stirred on a magnetic stirrer at 40 °C for 15 min. As a result, a chitosan and rosemary oil-based edible film coating solution was obtained. Similarly, for the chitosan and thyme oil combination, 2% (200 μL) thyme oil was added to the initially prepared chitosan solution. The mixture containing thyme oil was homogenized by stirring at 40 °C for 15 min with a magnetic stirrer, preparing the chitosan and thyme oil edible film coating solution.

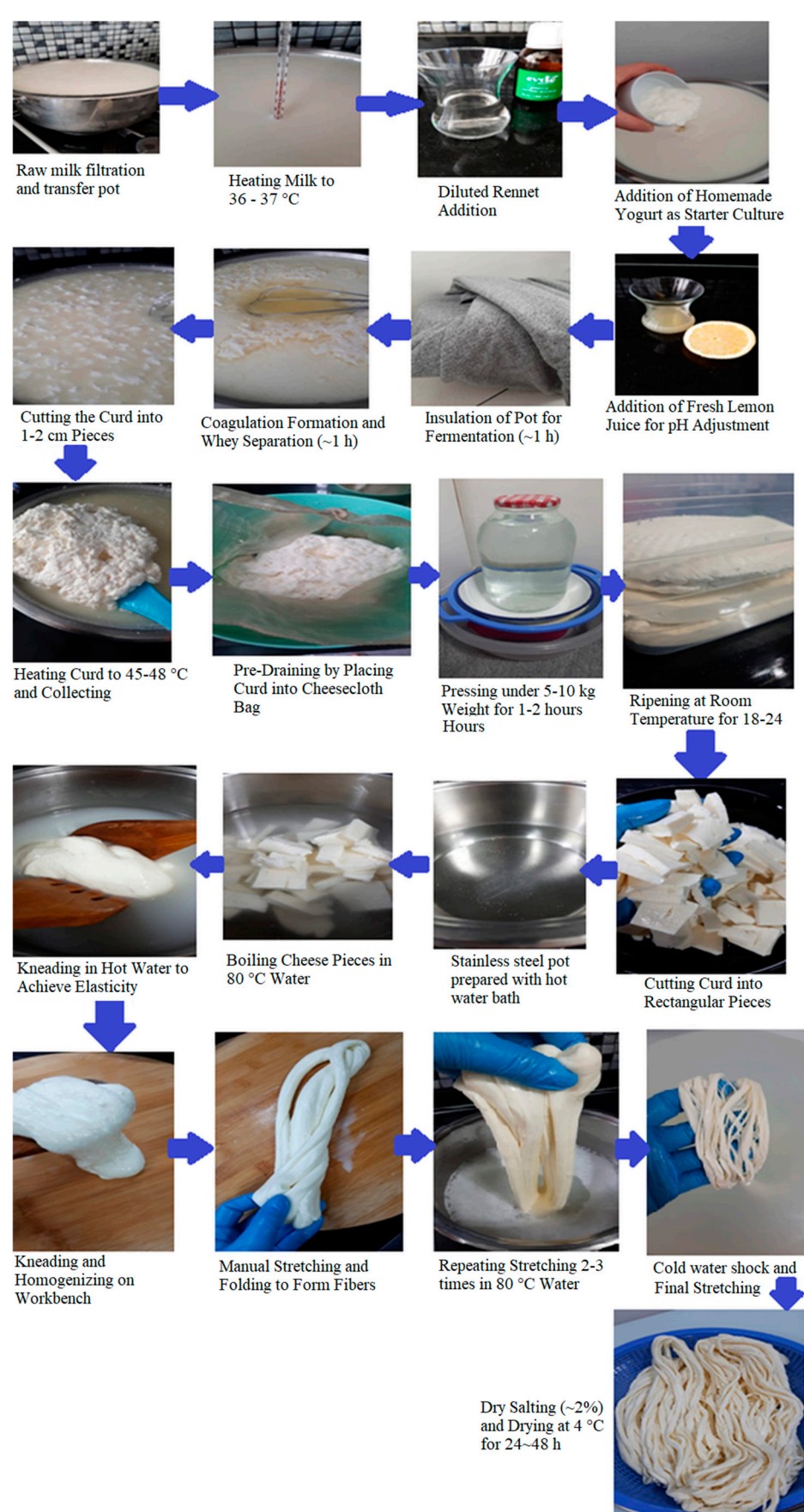

**Figure 2.** Traditional Cecil cheese production flowchart.

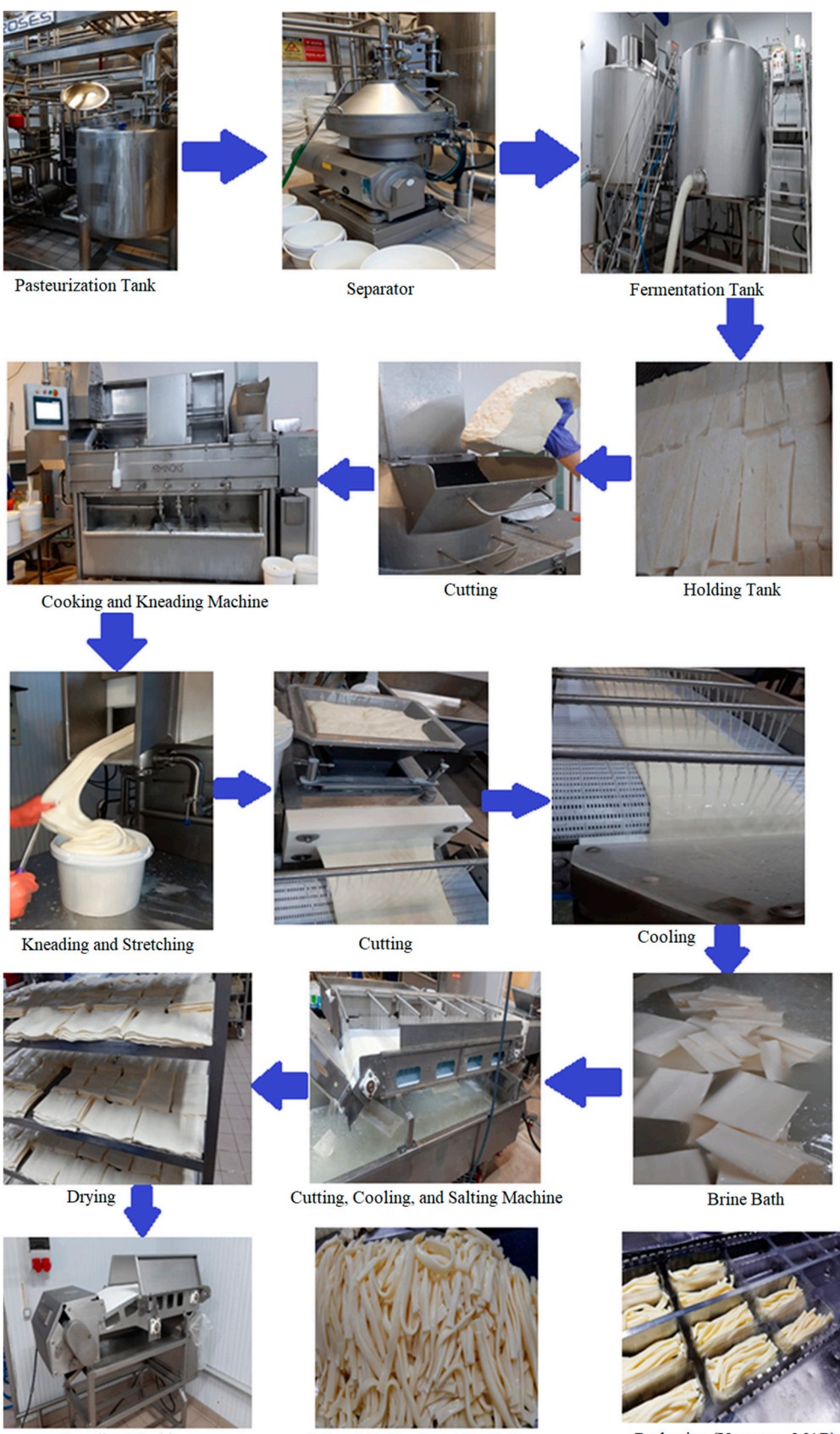

**Figure 3.** Industrial production flow chart.

Coating of Cecil Cheeses with Edible Films

The experimental study was based on two different production methods: traditionally produced Cecil cheese and industrially produced Cecil cheese. For both cheese groups, edible films containing chitosan, chitosan + thyme essential oil, and chitosan + rosemary essential oil were prepared, and coating procedures were performed using the dipping (immersion) method. For each production method, uncoated control groups without any edible film application were also established. Thus, a total of four groups were formed for each production method, defined as follows: industrial control group, industrial chitosan-coated group, industrial chitosan + thyme oil-coated group, and industrial chitosan + rosemary oil-coated group; traditional control group, traditional chitosan-coated group, traditional chitosan + thyme oil-coated group, and traditional chitosan + rosemary oil-coated group. After the coating process, all cheese samples were stored under appropriate cold storage conditions, and analyses were conducted at predetermined storage intervals. Examinations were carried out on days 1, 15, 30, and 45 of storage. At each time point, microbiological, physicochemical, and sensory analyses were performed, and the effects of edible film applications on the shelf life and quality parameters of Cecil cheese were evaluated. This experimental layout enabled a comprehensive comparative evaluation of the effects of coating composition and production method on the quality attributes of Cecil cheese (Figures 4 and 5).

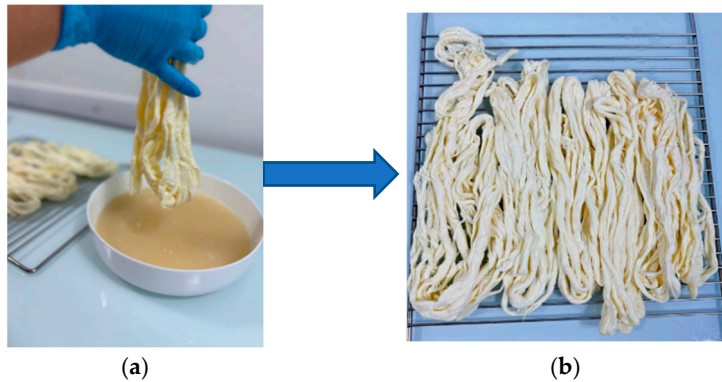

(**a**)      (**b**)

**Figure 4.** Coating of traditionally produced Cecil cheeses. (**a**) Dipping of Cecil cheese in chitosan (**b**) Drying of coated Cecil cheese.

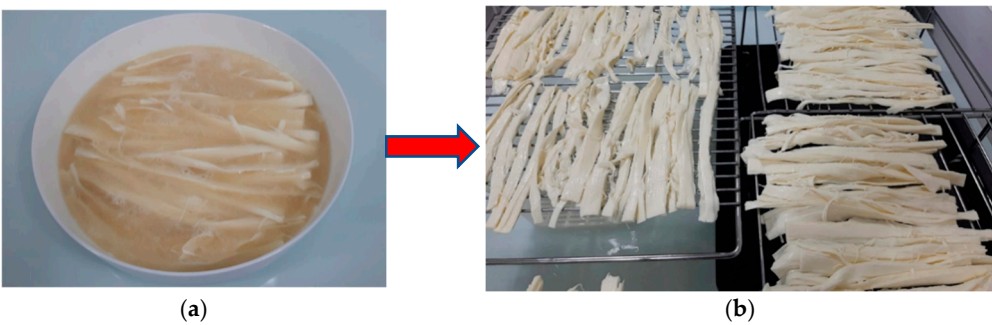

(**a**)      (**b**)

**Figure 5.** Coating of industrially produced Cecil cheeses. (**a**) Coating of industrial Cecil cheese (**b**) Drying of industrial Cecil cheese.

All cheese samples were coated using the dipping method to ensure uniform film application. Coating thickness was maintained as consistently as possible; however, minor variations during application may have occurred, potentially impacting microbiological and sensory results.

Storage and Analytical Procedure

After the coating process, all cheese samples were stored under appropriate cold storage conditions, and analyses were carried out at predetermined storage intervals. Examinations were conducted at four different time points: on the 1st, 15th, 30th, and 45th days of storage. At each time point, microbiological, physicochemical, and sensory analyses were performed to evaluate the effects of edible film applications on the shelf life and quality parameters of Cecil cheese.

*2.3. Analyses and Methods*

2.3.1. Antimicrobial Activity Analyses of Edible Films

The detection of foodborne pathogens is of critical importance for both public health and the sustainability of products such as milk and dairy products, which are quality, especially in products susceptible to microbial spoilage. In this context, *Salmonella* spp. and spore-forming species such as *Bacillus subtilis* are significant microorganisms in food microbiology.

In this thesis study, prior to the coating process, eight different film samples were prepared: chia, chia + thyme oil, chia + rosemary oil, chitosan, chitosan + thyme oil, chitosan + rosemary oil, chia + chitosan + thyme oil, and chia + chitosan + rosemary oil. The antimicrobial activity of each film was evaluated. Within this scope, the inhibitory potential of the film solutions against *Salmonella spp.* and *Bacillus subtilis* was analyzed.

The primary objective of these analyses was to determine the protective effectiveness of edible film materials before applying them to cheese and to identify which combinations were more effective from a microbiological perspective. Accordingly, only film combinations with proven antimicrobial effects were selected for use in subsequent cheese coating trials. The analyses were conducted following standard protocols widely used in the literature.

Detection of *Salmonella* spp.

For the detection of *Salmonella* spp., a pre-enrichment and selective enrichment protocol was implemented in accordance with ISO 6579-1:2017 [30]. Initially, 25 g of cheese sample was mixed with 225 mL of sterile Buffered Peptone Water (BPW; Merck) and incubated at 37 °C for 18–24 h. From the pre-enriched culture, 0.1 mL was transferred into tubes containing Rappaport-Vassiliadis Soya (RVS) Broth (Merck), and 1 mL was transferred into tubes containing Tetrathionate Broth (Merck). Samples were incubated at 42 °C for 24 h (RVS) and at 37 °C for 24 h (Tetrathionate), respectively. Following the selective enrichment step, streaking was performed on Xylose Lysine Deoxycholate Agar (XLD; Merck) and Brilliant Green Agar (BGA; Merck), and the plates were incubated at 37 °C for 24 h. After incubation, typical *Salmonella* colonies (pink-red colonies with black centers on XLD; red/pink-edged colonies on BGA) were selected and subjected to confirmation tests (urease, triple sugar iron agar, etc.) in accordance with ISO 6579-1:2017 [30].

Determination of *Bacillus subtilis*

For the enumeration of *Bacillus subtilis*, Tryptic Soy Agar (TSA; Merck) and appropriate dilution techniques were used. One milliliter from each of the serial dilutions ($10^{-1}$ to $10^{-7}$), prepared under sterile conditions, was transferred into sterile Petri dishes. TSA, cooled to approximately 45 °C, was poured over the samples using the pour plate method. After solidification, plates were incubated at 30 °C for 24 h. Subsequently, colonies morphologically resembling *Bacillus subtilis* were counted. Suspected colonies were confirmed via Gram staining, catalase testing, and spore staining [31].

Analyses of Cecil Cheese Coated with Edible Films

The quality changes in traditionally and industrially produced Cecil cheese were monitored during the storage period after edible film applications containing chitosan, chitosan + thyme oil, and chitosan + rosemary oil, chitosan combined with thyme oil, and chitosan combined with rosemary oil. Within this scope, all samples were stored under controlled conditions (+4 °C), and analyses were performed on days 1, 15, 30, and 45 of storage. At each time point, physical, chemical, microbiological, sensory, and textural analyses were conducted. Separate samples were taken for each analysis time, and all analyses were performed in triplicate. Key quality parameters such as color, pH, acidity, water activity, dry matter, texture, microbiological quality, and sensory evaluations were examined. Through these analyses, the effects of edible film applications on the shelf life and quality attributes of Cecil cheese were thoroughly assessed.

2.3.2. Physical Analyses of Cecil Cheese

This section describes the analyses conducted to determine the physical changes in Cecil cheese samples during storage. The physical analyses included the evaluation of color, water activity, and textural properties. These analyses allowed a detailed examination of the effects of edible film applications on appearance, structural integrity, and water activity as key quality parameters of Cecil cheese.

Color Analysis

Color measurements of Cecil cheese samples were performed according to the CIE L*, a*, b* color system. A PCE-brand color measurement device was used for this analysis. Prior to measurement, the device was calibrated using standard white and black calibration plates. Measurements were taken from three different points on each sample, with three repetitions at each point. The average values were used to determine L* (lightness), a* (redness-greenness), and b* (yellowness-blueness) parameters.

Water Activity ($a_w$) Analysis

The water activity ($a_w$) values of the Cecil cheese samples were determined using a Lab Swift Novasina brand water activity meter. Prior to each analysis, the device was calibrated with a calibration solution in accordance with the manufacturer's instructions. For the measurement, approximately 5 g of cheese sample was placed into the sample chamber of the device. During the measurement process, the device was operated in automatic stabilization mode to allow the sample to reach equilibrium. Each sample was measured in triplicate, and the average value was calculated and recorded for evaluation. Water activity analyses were performed on Days 1, 15, 30, and 45 of storage. These measurements were conducted to monitor the changes in water activity of the Cecil cheese samples throughout their shelf life.

2.3.3. Chemical Analyses of Cecil Cheese

To monitor the chemical changes occurring in Cecil cheese samples during storage, pH, titratable acidity, dry matter, and mineral content analyses were performed. These chemical analyses enabled the detailed observation of variations in acidity balance, pH stability, moisture and dry matter levels, and mineral composition within the cheese matrix. The primary aim was to assess the impact of edible film applications on the chemical quality of the cheese. All analyses were carried out in triplicate on storage days 1, 15, 30, and 45.

pH Analysis

The pH values of the Cecil cheese samples were determined using an HI 83141 model pH meter. Prior to measurement, the device was calibrated with standard buffer solutions

at pH 4.00 and pH 7.00 in accordance with the manufacturer's instructions. Approximately 10 g of each cheese sample was blended with 100 mL of distilled water to obtain a homogeneous mixture. The pH value was measured directly from the prepared mixture using a pH electrode. Each sample was analyzed in triplicate, and the average values were used for evaluation. The pH analyses were conducted on days 1, 15, 30, and 45 of storage to track chemical variations in the cheese samples over time.

Titratable Acidity Analysis

The titratable acidity of the Cecil cheese samples was determined according to the method described in the relevant literature [32]. For the analysis, 10 g of a homogenized cheese sample was weighed and ground in a mortar. Then, 100 mL of distilled water heated to 40 °C was then added to prepare the mixture. The resulting mixture was filtered using filter paper to obtain a clear filtrate. From this filtrate, 25 mL was taken, and 2–3 drops of 1% phenolphthalein solution were added. The samples were titrated with 0.1 N NaOH solution until a light pink color appeared. The titratable acidity was then calculated as a percentage of lactic acid based on the amount of NaOH used. Acidity analyses were carried out on days 1, 15, 30, and 45 of storage.

Dry Matter Analysis

The dry matter content of the Cecil cheese samples was determined using a RADWAG MA 50.R model rapid moisture analyzer. During the measurements, the moisture determination mode of the device was selected. Approximately 4 g of each sample was weighed and placed into the sample chamber of the device. The device automatically detected the moisture content in the sample, and the results were obtained directly as a percentage of dry matter (% DM). Each sample was analyzed once. Dry matter analyses were performed on storage days 1, 15, 30, and 45.

2.3.4. Microbiological Analyses of Cecil Cheese

Microbiological analyses were performed on Cecil cheese samples to evaluate the microbiological quality and shelf life during storage. Counts were conducted for total aerobic mesophilic bacteria, yeast and mold, coliform bacteria, and lactic acid bacteria. These analyses aimed to determine the effects of edible film coatings on microbiological control and the rate of microbial growth in cheeses produced by different methods. Microbiological analyses were performed on days 1, 15, 30, and 45 of storage, and each analysis was carried out in triplicate. The results were expressed as logarithmic colony-forming units per gram (log CFU/g). Before microbiological analysis, dilution solutions containing 0.85% NaCl were prepared and sterilized by autoclaving at 121 °C for 15 min. Subsequently, 99 mL of sterile dilution solution was placed into sterile jars, and 11 g of cheese samples were weighed under sterile conditions and transferred to the jars. Samples were then homogenized using a Stomacher for 1 min. This process yielded the initial $10^{-1}$ dilutions. Further serial dilutions up to $10^{-7}$ were prepared using sterile pipettes by transferring 1 mL from each previous dilution into tubes containing 9 mL of sterile dilution solution [33].

Total Aerobic Mesophilic Bacteria Count

Plate Count Agar (PCA; Merck) medium was used for total aerobic mesophilic bacteria (TAMB) counts. The pour plate method was employed. From appropriate dilutions, 1 mL of the sample was transferred into duplicate Petri dishes, and approximately 15 mL of PCA medium cooled to 45 °C, was added to each dish. After ensuring homogeneous mixing, the plates were left to solidify and then incubated at 30–32 °C for 48 h. After incubation, colonies were counted, and results were calculated as log CFU/g [32].

Yeast and Mold Count

Dichloran Rose Bengal Chloramphenicol (DRBC) Agar (Lab 217, Lancashire, UK) was used to determine yeast and mold counts. The surface spread method was applied using appropriate dilutions. The inoculated Petri dishes were incubated at $25 \pm 1\,°C$ for 5 days. After incubation, yeast and mold colonies were counted, and the results were expressed as log CFU/g [34].

Coliform Bacteria Count

Violet Red Bile Agar (VRBA; Merck) was used for coliform bacteria counts in Cecil cheese samples. From each appropriate dilution, 1 mL was transferred into duplicate Petri dishes, and then 13–15 mL of VRBA medium cooled to $45\,°C$ was added. After mixing, plates were inverted and incubated at $35 \pm 2\,°C$ for 48 h. Colonies that were pink or red with a diameter greater than 0.5 mm were considered coliforms and counted. Results were expressed as log CFU/g [32].

Lactic Acid Bacteria Count

The Man, Rogosa, Sharpe Agar (MRS Agar; Merck) was used to determine lactic acid bacteria counts. After inoculating the medium with appropriate dilutions, the Petri dishes were incubated under anaerobic conditions at $35\,°C$ for 36 h. After the initial incubation, the plates were further incubated aerobically at $35\,°C$ for an additional 36 h. Colonies that were catalase-negative and Gram-positive with typical morphology were counted. Results were expressed as log CFU/g [35].

2.3.5. Sensory Analysis of Cecil Cheese

Sensory analyses were conducted to evaluate changes in the sensory properties of Cecil cheese during storage after edible film coating. The sensory panel consisted of 20 untrained assessors, randomly selected from the staff and students of [Amasya University Suluova Vocational School], with no known allergies to dairy products or sensory impairments. Different panelists participated in each analysis day to prevent bias from repeated exposure. Evaluations were performed on storage days 1, 15, 30, and 45 in a sensory laboratory equipped with individual booths under D65 natural white lighting. Prior to evaluation, cheese samples (approximately 10 g) were equilibrated to $12 \pm 1\,°C$ and served on odorless white plates coded with three-digit random numbers. Drinking water and unsalted crackers were provided as palate cleansers between samples. Evaluation criteria included color and appearance, texture, odor, flavor-aroma, saltiness, and overall acceptability. A 9-point hedonic scale was used, where 1 represented "extremely poor" and 9 represented "excellent." In addition, panelists were asked, *"Would you purchase such a product if it were available on the market?"* They responded using a 3-point scale (1 = definitely would buy, 2 = might buy, 3 = definitely would not buy). All sensory evaluations were conducted in accordance with ISO 21527-1 (ISO, 2008).standards. All participants were fully informed prior to the analysis and voluntarily agreed to participate. Sensory data were analyzed based on the mean scores of the panelists, as described by [36].

2.3.6. Texture Profile Analysis (TPA)

The textural properties of Cecil cheese samples were evaluated using a Texture Analyzer. A standard P/5 cylindrical probe was used for the measurements. Each sample was prepared as a cylinder approximately 2 cm in length and 1 cm in diameter. During the analysis, a two-cycle compression at a speed of 1 mm/s was applied to the samples, and the maximum force (N) and compression distance were recorded. These measurements were used to determine the hardness (N) of the cheese. Each sample was tested in triplicate,

and average values were used for evaluation. Texture analyses were performed on storage days 1 and 15 to assess structural durability [37].

### 2.3.7. Statistical Analysis

All analyses were performed in at least two replicates, and the results were presented as mean ± standard deviation (SD). All results were expressed as mean ± standard deviation calculated using SPSS (SPSS Statistics 17.0, Armonk, NY, USA). To determine whether the effects of cheese type and storage period on the physicochemical, microbiological, sensory, and textural properties of Cecil cheese samples were statistically significant, a two-way analysis of variance (two-way ANOVA) was performed using the JMP software (version 5.0.1a; SAS Institute, Inc., Cary, NC, USA) at a significance level of $\alpha = 0.05$.

## 3. Results and Discussions

### 3.1. Evaluation of Antimicrobial Activity Results of Edible Films

In this study, eight different edible film samples were tested in vitro for their antimicrobial effects against *Salmonella* and *Bacillus subtilis* bacteria. The antimicrobial activity was evaluated based on the diameter (mm) of the inhibition zones formed. Only the film sample containing solely chitosan exhibited significant inhibition against both microorganisms, with inhibition zone diameters of 23 mm for *Salmonella* and 21 mm for *B. subtilis* (Table 2). This result demonstrates that the antimicrobial properties of chitosan alone are highly effective. The positively charged structure of chitosan interacts electrostatically with the negatively charged components of the microbial cell membrane, disrupting cell permeability, which explains the antimicrobial effect [38,39].

**Table 2.** Antimicrobial activity analysis results of edible films (inhibition zone diameter, mm).

| Edible Film | *Salmonella* | *Bacillus subtilis* |
|---|---|---|
| Chia seed | – | – |
| Chitosan | 23 mm | 21 mm |
| Chia seed + Rosemary oil | – | – |
| Chia seed + Thyme oil | – | – |
| Chitosan + Rosemary oil | 16 mm | – |
| Chitosan + Thyme oil | 32 mm | 33 mm |
| Chia seed + Chitosan + Rosemary oil | 20 mm | 16 mm |
| Chia seed + Chitosan + Thyme oil | 22 mm | 23 mm |

The highest antimicrobial activity was observed with the chitosan + thyme essential oil combination. This formulation produced inhibition zones measuring 32 mm against *Salmonella* and 33 mm against *B. subtilis*. Thyme essential oil enhances antimicrobial efficacy by causing disruptions in the cell membrane due to its phenolic compounds such as carvacrol and thymol, exhibiting a synergistic effect when combined with chitosan [40,41]. The chitosan + rosemary essential oil combination showed inhibition only against *Salmonella* with a zone diameter of 16 mm, while no effect was observed against *B. subtilis,* suggesting a more limited spectrum of activity.

Notably, films containing chia alone or combined with essential oils showed no inhibition, while chia and chitosan combinations demonstrated moderate activity. The chia + chitosan + thyme essential oil formulation produced inhibition zones of 22 mm and 23 mm, indicating maintained synergistic effects despite the presence of chia.

The concentration of essential oils in chitosan-based films is critical for balancing antimicrobial efficacy with sensory quality. Higher essential oil content generally enhances antimicrobial activity; however, elevated concentrations—particularly of rosemary oil—can

reduce aroma and taste scores. Optimized, lower concentrations of thyme oil provide effective antimicrobial protection while maintaining acceptable sensory properties. Moreover, the literature reports that chitosan-essential oil combinations can generate specific interactions within the polymer matrix, influencing both antimicrobial efficacy and controlled release of the oils [42]. These interactions enable sustained microbial inhibition on the cheese surface, while high essential oil levels may negatively impact sensory attributes. Therefore, careful optimization of essential oil concentration is essential to maximize functional performance without compromising consumer acceptance.

This suggests that the antimicrobial activity of rosemary oil may be more limited compared to thyme oil. The literature also reports that the antimicrobial effect of rosemary oil varies depending on the target microorganism and the concentration used [43].

Film groups containing chia (chia, chia + thyme essential oil, and chia + rosemary essential oil) did not exhibit inhibition against either bacterium. This finding indicates that the polysaccharide-based structures derived from chia seeds are insufficient in terms of antimicrobial activity and do not show efficacy alone or in combination with thyme or rosemary oils. However, chia combined with chitosan demonstrated moderate activity. Notably, the chia + chitosan + thyme essential oil formulation produced inhibition zones of 22 mm and 23 mm against *Salmonella* and *B. subtilis*, respectively. These values suggest that the synergistic effect is maintained despite the presence of chia and that the combination of chitosan with thyme essential oil remains effective.

In conclusion, the most effective formulations in terms of antimicrobial activity were chitosan + thyme essential oil, chitosan alone, and chia + chitosan + thyme essential oil, respectively. Conversely, groups containing chia and essential oils but lacking chitosan were ineffective. In light of these data, it was decided that only the formulations demonstrating antimicrobial efficacy—chitosan, chitosan + thyme essential oil, and chitosan + rosemary essential oil—would be used in the subsequent phases of this study.

### 3.1.1. Evaluation of the Physicochemical Analysis Results of Cecil Cheese Samples

Water activity ($a_w$) is one of the most critical parameters for the microbial stability and shelf life of a food product. In this study, the effects of different coating applications on the $a_w$ of Cecil cheese samples produced by traditional and industrial methods were evaluated on days 1, 15, 30, and 45. On the first day, the $a_w$ values across all groups ranged between 0.87 and 0.94. The highest $a_w$ value (0.94) was detected in the IEK and IBR groups, which correspond to industrial production samples coated with chitosan combined with thyme essential oil and rosemary essential oil, respectively. The lowest $a_w$ value (0.87) was observed in the IK group (industrial control group). As the storage period progressed, a general decline in $a_w$ values was observed across all samples. This decrease can be attributed to the reduction in free water and the development of a more compact structure during the cheese ripening process. Examination of the data on day 45 revealed that the lowest $a_w$ value (0.82) was recorded in the TG (traditional control group) sample, indicating that the uncoated traditional cheese had the lowest free water content. On the same day, the highest $a_w$ values (0.87) were noted in the TBR and TEK groups. This finding points to the moisture-retaining effect of chitosan coatings containing thyme and rosemary essential oils. Samples coated with films containing chitosan and essential oils generally maintained higher $a_w$ values throughout the storage period compared to control groups. This suggests that edible films can limit moisture transfer, thereby stabilizing the water activity of the product. In particular, $a_w$ values of cheeses produced industrially and coated with chitosan and essential oil were found to be statistically significantly higher than those produced traditionally ($p < 0.05$) (Table 3). In a study conducted by Yüceer (2017), chitosan-based films applied to Cecil cheese were reported to stabilize moisture content and thus water

activity [44]. Similarly, Karakuş (2021) noted that chitosan-containing coatings limited yeast and mold growth and prevented decreases in water activity [45]. These findings are consistent with the observations of the present study. The high moisture retention capacity of films containing thyme essential oil is also compatible with their reported antimicrobial effects in the literature [40,46]. The observed $a_w$ value ranges in this study align with those reported in previous research. The characteristic structure of Cecil cheese and the storage duration. For instance, Özkan (2018) reported $a_w$ values around 0.85 in traditionally produced Cecil cheeses and noted that this value influences the cheese's susceptibility to moisture loss [46]. From a water activity perspective, cheese samples coated with chitosan and essential oils exhibited a more stable structure compared to control groups. It is thought that essential oil-enriched chitosan films both reduce moisture loss and suppress microbial growth, thereby enhancing the product's shelf life. The higher $a_w$ values observed in industrially produced samples compared to traditional ones may be related to more stable process control. Differences observed between traditional and industrial samples can be attributed to variations in cheese structure and production procedures. Cheeses produced through industrial methods generally exhibit a more homogeneous structure and form denser cheese matrices, which in turn limits moisture evaporation. Similarly, Atarés and Chiralt (2016) reported that the combination of hydrophobic lipids (such as essential oils) and hydrophilic biopolymers (such as chitosan) creates a synergistic barrier against water loss [11]. Our findings are also consistent with previous studies in which the application of chitosan-thyme film in Kashar cheese and alginate-rosemary film in white cheese significantly reduced weight loss [47,48]. The obtained water activity values are in agreement with the literature and demonstrate that edible films containing essential oils provide effective moisture retention. The observed differences between traditional and industrial samples may be attributed to factors such as production methods, the matrix structure of the cheese, and the adhesion efficiency of the edible coatings.

When examining the pH values of Cecil cheese samples included in this study, a significant decreasing trend was generally observed across all groups as the storage period progressed ($p < 0.05$). This decline can be attributed to microbial activity and lactic acid formation occurring during the cheese ripening process. In terms of initial pH values, the highest pH was recorded in the EKE group (industrial production, chitosan + thyme oil) with a value of 6.32. This was followed by the EBİ (6.30), GKİ (6.31), and GBİ (6.30) groups. The lowest initial pH value was measured in the traditional control group (GK) at 5.78. This suggests that edible film coatings may have an elevating effect on initial pH values (Table 3). In particular, films containing essential oils (thyme and rosemary) may have maintained higher pH by limiting the release of acidic compounds from the film matrix.

By the end of the storage period (day 45), the lowest pH was observed in the GKİ group (5.20), followed by the GK (5.26), EKİ (5.28), and EK (5.30) groups. The highest final pH value was recorded in the GBİ group at 5.37. These results indicate that films containing chitosan and essential oils may partially suppress the natural development of acidity in cheese. Specifically, the combination of chitosan and rosemary oil in the traditional production group (GBİ) appeared to limit pH reduction, thereby contributing to product stability. Accordingly, there were statistically significant differences between the GK group and all other coated groups in terms of initial pH values ($p < 0.05$). This may be due to the absence of a protective layer in the traditional control group, resulting in faster acidification. Similar pH ranges have been reported for Cecil cheese in previous studies. For instance, Karakuş (2021) reported pH values between 5.25 and 6.20 in Cecil cheese samples subjected to different protective applications [45]. Likewise, Yüceer (2017) investigated the effect of film coatings on pH and emphasized that films containing thyme oil slowed the decrease in acidity [44]. These findings are consistent with the results of the

present study. Dikbaş et al. (2010) reported pH values ranging from 4.92 to 5.67 in Cecil cheese samples collected from the Erzurum region [49]. Similar pH values were observed in the control group samples of our study. Additionally, Çağrı et al. (2004) reported that pH reduction occurred more slowly in dairy products coated with chitosan films, attributing this effect to the antimicrobial activity of chitosan [50]. This supports our observation that pH decline was more limited in coated samples compared to the control group. However, changes in pH values are directly related to both the composition of the coatings and the production method. The combination of chitosan and essential oils was found to provide more effective stabilization, particularly in industrially produced samples.

The slower pH decline in coated samples can be attributed to the **multifunctional properties of chitosan and essential oils**. Chitosan forms a physical barrier on the cheese surface due to its polymeric structure, limiting microbial contact and directly inhibiting microbial growth. Essential oils (rosemary and thyme) contain antimicrobial compounds that specifically restrict the metabolic activity of lactic acid bacteria, thereby reducing acid formation. This combination allows the coating to **physically and chemically control acid development and stabilize pH** on the cheese surface. In industrially produced samples, the effect is more pronounced due to the homogeneous distribution and stronger adhesion of the coating. Additionally, the essential oil and chitosan components regulate water activity and surface moisture of the cheese, further limiting microbial growth. These observations align with similar findings in the literature. For instance, Çağrı et al. (2004) and Yüceer (2017) reported that chitosan and essential oil-containing coatings reduced microbial activity and slowed pH decline [44,50]. Furthermore, Iqbal et al. (2021) and Ressutte et al. (2022) demonstrated that pH stabilization is also related to the buffering capacity of coatings [51,52]. Collectively, these findings support the idea that active coatings, particularly those enriched with essential oils, slow pH reduction and contribute to maintaining both microbial safety and flavor balance in cheese.

Our findings are in alignment with those of Karakuş (2021), who reported similar pH variations in coated cheese samples [45]. Yüceer (2017) also observed that films containing thyme essential oil slowed the development of acidity, which was attributed to the antimicrobial effects on lactic acid bacteria [44]. Similarly, Çağrı et al. (2004) determined that chitosan coatings reduced microbial growth in dairy products, thereby contributing to pH stability [50]. Iqbal et al. (2021) and Ressutte et al. (2022) observed that films containing citric acid possessed buffering capacity, which helped to stabilize pH levels [51,52]. The observed pH differences between coated and uncoated samples also indicate that active coatings effectively suppress undesirable microbial activity. Casalini et al. (2024) and El-Sayed and Youssef (2024) similarly reported that active coatings limited the growth of lactic acid bacteria, contributing to the maintenance of higher pH values [53,54]. In our study, higher pH values were particularly detected in industrially produced samples coated with essential oil-enriched films. This observation is consistent with the literature. Therefore, the data obtained in this study are in agreement with previous research. It was determined that active coatings, especially those containing essential oils, reduced the decrease in pH during cheese ripening. This contributes to the preservation of flavor balance and microbial safety.

The titratable acidity of the Cecil cheese samples examined in this study showed significant variations depending on the storage period and the applied edible film formulations ($p < 0.05$). Overall, acidity values increased over time, which can be considered an indicator of microbial fermentation and, particularly, the activity of lactic acid bacteria. On day 1, the lowest acidity value was observed in the EKE sample (industrial production, chitosan + thyme oil) at $1.05 \pm 0.02$, while the highest value was recorded in the GKE group (traditional production, chitosan + thyme oil) at $1.18 \pm 0.01$. This finding suggests that the

same formulation may yield different results depending on the production method. By day 15 of storage, differences among samples became more pronounced. The acidity levels in the GBİ and EKİ samples were measured at 1.32 ± 0.02 and 1.30 ± 0.02, respectively (Table 3). These results suggest that, in addition to chitosan content, the presence of essential oils may have limited microbial growth to some extent, thereby allowing for a more controlled increase in acidity.

**Table 3.** Physicochemical properties of Cecil cheese samples during storage.

| Parameter | Sample Code | Storage Days | | | |
|---|---|---|---|---|---|
| | | Day 1 | Day 15 | Day 30 | Day 45 |
| **Water activity (aw)** | GK | 0.88 $^{Db}$ ± 0.01 | 0.92 $^{Ba}$ ± 0.02 | 0.84 $^{Fc}$ ± 0.01 | 0.82 $^{Ec}$ ± 0.01 |
| | GKİ | 0.88 $^{Db}$ ± 0.01 | 0.93 $^{Aa}$ ± 0.01 | 0.87 $^{Db}$ ± 0.01 | 0.80 $^{Dc}$ ± 0.01 |
| | GBİ | 0.88 $^{Cb}$ ± 0.01 | 0.90 $^{CDa}$ ± 0.01 | 0.87 $^{Dc}$ ± 0.01 | 0.87 $^{Ac}$ ± 0.01 |
| | GKE | 0.89 $^{Cab}$ ± 0.01 | 0.93 $^{Ca}$ ± 0.05 | 0.88 $^{Cab}$ ± 0.01 | 0.87 $^{Ab}$ ± 0.01 |
| | EK | 0.87 $^{Da}$ ± 0.01 | 0.87 $^{Da}$ ± 0.01 | 0.86 $^{Eb}$ ± 0.01 | 0.86 $^{Cb}$ ± 0.01 |
| | EKİ | 0.93 $^{Ba}$ ± 0.01 | 0.89 $^{CDb}$ ± 0.01 | 0.88 $^{Bc}$ ± 0.01 | 0.87 $^{ABd}$ ± 0.01 |
| | EBİ | 0.94 $^{Aa}$ ± 0.01 | 0.88 $^{Dc}$ ± 0.01 | 0.89 $^{Ab}$ ± 0.01 | 0.86 $^{BCd}$ ± 0.04 |
| | EKE | 0.94 $^{Aa}$ ± 0.01 | 0.89 $^{CDb}$ ± 0.01 | 0.89 $^{Ab}$ ± 0.01 | 0.86 $^{BCc}$ ± 0.04 |
| **pH** | G K | 5.78 $^{Ea}$ ± 0.02 | 5.67 $^{Fa}$ ± 0.04 | 5.43 $^{Cb}$ ± 0.10 | 5.26 $^{CDc}$ ± 0.06 |
| | GKİ | 6.31 $^{Aa}$ ± 0.02 | 5.66 $^{Fb}$ ± 0.04 | 5.47 $^{Cc}$ ± 0.01 | 5.20 $^{Dd}$ ± 0.05 |
| | GBİ | 6.30 $^{Aa}$ ± 0.01 | 5.88 $^{Db}$ ± 0.03 | 5.64 $^{ABc}$ ± 0.04 | 5.37 $^{Ad}$ ± 0.03 |
| | GKE | 5.96 $^{Da}$ ± 0.02 | 5.73 $^{Eb}$ ± 0.03 | 5.64 $^{ABc}$ ± 0.01 | 5.34 $^{ABd}$ ± 0.01 |
| | EK | 6.01 $^{Ca}$ ± 0.03 | 5.94 $^{Cb}$ ± 0.01 | 5.57 $^{Bc}$ ± 0.01 | 5.30 $^{ABCd}$ ± 0.04 |
| | EKİ | 6.16 $^{Ba}$ ± 0.05 | 6.09 $^{Bb}$ ± 0.01 | 5.58 $^{ABc}$ ± 0.01 | 5.28 $^{BCd}$ ± 0.01 |
| | EBİ | 6.30 $^{Aa}$ ± 0.01 | 6.04 $^{Bb}$ ± 0.01 | 5.61 $^{ABc}$ ± 0.01 | 5.32 $^{ABCd}$ ± 0.02 |
| | EKE | 6.32 $^{Aa}$ ± 0.01 | 6.19 $^{Ab}$ ± 0.01 | 5.65 $^{Ac}$ ± 0.01 | 5.36 $^{Ad}$ ± 0.04 |
| **Titratable acidity (% lactic acid)** | G K | 1.14 $^{Bd}$ ± 0.02 | 1.23 $^{Dc}$ ± 0.01 | 1.39 $^{BCb}$ ± 0.05 | 1.66 $^{Ba}$ ± 0.03 |
| | GKİ | 1.09 $^{CDd}$ ± 0.01 | 1.25 $^{BCDc}$ ± 0.02 | 1.37 $^{Cb}$ ± 0.01 | 1.57 $^{Ca}$ ± 0.01 |
| | GBİ | 1.16 $^{ABd}$ ± 0.02 | 1.32 $^{Ac}$ ± 0.02 | 1.44 $^{ABCb}$ ± 0.02 | 1.66 $^{Ba}$ ± 0.01 |
| | GKE | 1.18 $^{Ac}$ ± 0.01 | 1.29 $^{ABCb}$ ± 0.06 | 1.37 $^{Cb}$ ± 0.07 | 1.67 $^{Ba}$ ± 0.03 |
| | EK | 1.09 $^{CDd}$ ± 0.01 | 1.27 $^{ABCDc}$ ± 0.01 | 1.39 $^{BCb}$ ± 0.04 | 1.61 $^{Ca}$ ± 0.02 |
| | EKİ | 1.10 $^{Cd}$ ± 0.02 | 1.30 $^{ABc}$ ± 0.02 | 1.50 $^{Ab}$ ± 0.03 | 1.78 $^{Aa}$ ± 0.01 |
| | EBİ | 1.06 $^{DEd}$ ± 0.01 | 1.27 $^{ABCDc}$ ± 0.01 | 1.47 $^{ABb}$ ± 0.06 | 1.74 $^{Aa}$ ± 0.01 |
| | EKE | 1.05 $^{Ed}$ ± 0.02 | 1.24 $^{CDc}$ ± 0.03 | 1.45 $^{ABCb}$ ± 0.03 | 1.74 $^{Aa}$ ± 0.04 |
| **Dry matter (%)** | GK | 51.59 $^{BCd}$ ± 0.53 | 52.80 $^{Dc}$ ± 0.30 | 55.27 $^{Bb}$ ± 0.76 | 56.64 $^{Aa}$ ± 0.29 |
| | GKİ | 49.95 $^{Dd}$ ± 0.05 | 51.46 $^{Ec}$ ± 0.29 | 53.57 $^{Cb}$ ± 0.34 | 55.44 $^{Ca}$ ± 0.43 |
| | GBİ | 50.81 $^{CDc}$ ± 1.70 | 53.13 $^{CDb}$ ± 0.12 | 55.70 $^{ABa}$ ± 0.21 | 56.65 $^{Aa}$ ± 0.15 |
| | GKE | 51.88 $^{BCc}$ ± 0.61 | 53.39 $^{Cb}$ ± 0.36 | 55.88 $^{ABa}$ ± 0.15 | 56.41 $^{Aa}$ ± 0.03 |
| | EK | 52.48 $^{ABc}$ ± 0.14 | 54.81 $^{Bb}$ ± 0.17 | 56.17 $^{Aa}$ ± 0.21 | 56.37 $^{ABa}$ ± 0.51 |
| | EKİ | 53.29 $^{Ad}$ ± 0.18 | 54.61 $^{Bc}$ ± 0.27 | 55.23 $^{Bb}$ ± 0.12 | 55.90 $^{BCa}$ ± 0.11 |
| | EBİ | 53.54 $^{Ac}$ ± 0.23 | 55.73 $^{Ab}$ ± 0.28 | 56.37 $^{Aa}$ ± 0.40 | 56.89 $^{Aa}$ ± 0.13 |
| | EKE | 52.80 $^{ABd}$ ± 0.09 | 54.62 $^{Bc}$ ± 0.43 | 55.86 $^{ABb}$ ± 0.14 | 56.48 $^{Aa}$ ± 0.18 |

Mean ± standard deviation. **Note:** Values are expressed as **mean ± standard deviation**. **A–F:** Capital letters in the same column indicate statistically significant differences between sample types on the same storage day (*p* < 0.05). **a–d:** Lowercase letters in the same row indicate statistically significant differences within the same sample over different storage times (*p* < 0.05). **Sample Codes: GK:** Traditional production, control group, **GKİ:** Traditional production, Cecil cheese coated with chitosan, **GBİ:** Traditional production, Cecil cheese coated with chitosan + rosemary essential oil, **GKE:** Traditional production, Cecil cheese coated with chitosan + thyme essential oil, **EK:** Industrial production, control group, **EKİ:** Industrial production, Cecil cheese coated with chitosan, **EBİ:** Industrial production, Cecil cheese coated with chitosan + rosemary essential oil, **EKE:** Industrial production, Cecil cheese coated with chitosan + thyme essential oil.

On day 30, the EKİ sample reached the highest acidity level at $1.50 \pm 0.03$, while the lowest was recorded in the GKİ group at $1.37 \pm 0.01$. These results imply that when chitosan is used alone, it may offer a more stable acidity profile compared to combinations with essential oils. By day 45, the highest acidity value was observed in the EKİ group ($1.78 \pm 0.01$), while the lowest was recorded in the GKİ sample ($1.57 \pm 0.01$). The film combinations applied in industrial production had a greater impact on acidity accumulation compared to traditional production. On the same day, EBİ and EKE samples also exhibited high acidity values, reaching $1.74 \pm 0.01$ and $1.74 \pm 0.04$, respectively. This indicates that while the addition of essential oils may limit microbial activity, it does not completely suppress lactic acid production. The findings of this study are consistent with those reported by Karakuş (2021), who found that combinations of chitosan and essential oils slowed the increase in acidity and ensured more stable values throughout storage in a study on Kashar cheese [45]. Similarly, Yüceer (2017) emphasized that thyme oil supplementation limited microbial load in cheese and prevented sudden increases in acidity [44]. Moreover, some studies have reported that due to the strong antimicrobial properties of essential oils, acidity levels remained significantly lower. For example, Arfat et al. (2015) demonstrated that edible films containing thyme oil suppressed fermentative activity in hard cheeses and significantly limited the increase in acidity [54]. Bleoancă et al. (2020) found that the application of essential oil-based films on hard cheeses inhibited fermentative activity [55]. Similarly, Kavas et al. (2015) and Karagöz and Demirdöven (2019) reported that alginate and chitosan coatings slowed the development of acidity, which contributed to improved sensory acceptance [47,48]. Dikbaş et al. (2010) reported acidity values ranging from 0.32 to 1.05 in Cecil cheese samples collected from the Erzurum region [49]. Similar acidity values were observed in the control group samples of our study. In the present study, chitosan alone provided more stable acidity values, while combinations of essential oils resulted in greater variability. This may be due to partial inhibition of microbial fermentation. Differences in the cheese matrix could also contribute to this outcome. Variations between samples produced by traditional and industrial methods may be attributed to differences in the structural and microbial characteristics of the cheese, as well as differences in the adhesion efficiency of the films. In summary, our findings suggest that edible films can stabilize acidity increase during storage. Furthermore, the use of essential oils offers additional protection against rapid lactic acid accumulation.

On the first day of storage, the highest dry matter content was determined in the GKE group (traditional, chitosan + thyme oil) at $51.88 \pm 0.61\%$, followed by the traditional control group (GK) with $51.59 \pm 0.53\%$. The lowest value was observed in the GKİ sample (traditional, containing only chitosan) at $49.95 \pm 0.05\%$. This suggests that films containing thyme oil may have a limited moisture retention capacity, whereas the application of chitosan alone may better reduce moisture loss. As the storage period progressed, an increase in dry matter content was observed in all samples. By day 45, the dry matter content in the GK group rose to $56.64 \pm 0.29\%$, while the GKE and GBİ groups reached $56.41 \pm 0.03\%$ and $56.65 \pm 0.15\%$, respectively (Table 3). These increases indicate ongoing moisture loss during storage and varied impacts depending on the films' water vapor permeability. The lowest increase was again seen in the GKİ sample, with a value limited to $55.44 \pm 0.43\%$, suggesting that chitosan-containing films exhibit greater moisture retention properties.

Dikbaş et al. (2010) reported dry matter content ranging from 34.8% to 59.3% in Cecil cheese samples collected from the Erzurum region [49]. Similar dry matter values were observed in the control group samples of our study. When compared to other studies in the literature, Karakuş (2021) reported a dry matter content of around 50% on day 1 and rising to approximately 56% by day 45 in uncoated Cecil cheese [45]. This is consistent with our findings and confirms the impact of film permeability on dry matter content.

Similarly, Yüceer (2017) reported that the combination of chitosan and thyme oil was effective in moisture control in Kashar cheese, observing more balanced dry matter increases throughout storage [44]. These findings align with the high dry matter values observed in the GKE group both at the start and the end of storage in our study.

In conclusion, our study demonstrated that edible film coatings directly affect the dry matter content of Cecil cheese, with thyme oil-containing films causing faster moisture loss. However, this moisture loss may reduce the risk of microbial growth, potentially extending shelf life. However, chitosan-containing films provide a more stable moisture structure, helping to maintain the cheese's textural softness and enhancing consumer acceptance. The data obtained largely coincide with the literature findings, with some variations attributable to differences in film composition, application methods, and the initial cheese composition. The results of this study are consistent with Karakuş (2021), who observed similar increases in dry matter in uncoated cheeses [45]. Similarly, Yüceer (2017) reported that coatings containing thyme oil provided better moisture control, aligning with our findings [44]. Artiga-Artigas et al. (2017) and Molina-Hernández et al. (2020) demonstrated that thyme oil-based coatings reduce moisture loss in cheeses [21,56]. Conversely, Pires et al. (2024) indicated that such coatings may increase moisture content depending on their permeability properties [57]. Coatings containing thyme oil allowed slightly greater moisture loss compared to films containing only chitosan. This phenomenon may be attributed to the modification of the film matrix structure and barrier properties by essential oils. However, this moisture loss could be beneficial in terms of microbial safety. Overall, the obtained results are in agreement with the literature, and minor differences may arise from variations in the initial composition of the cheese, film formulations, and storage conditions.

The decrease in water activity and the concurrent increase in dry matter content during storage were closely associated with changes in the textural properties of Cecil cheese. Hardness, chewiness, and gumminess were better preserved in samples coated with edible films, indicating that moisture retention played a key role in stabilizing texture. Coatings containing chitosan and essential oils effectively mitigated moisture loss and maintained structural integrity, and while partially suppressing acidification, contributed to stabilizing elasticity and adhesiveness by slowing proteolysis-related changes. Industrial cheeses, characterized by a denser and more homogeneous matrix, exhibited higher moisture retention and better preservation of hardness and chewiness compared to traditionally produced cheeses. Furthermore, films combining chitosan with essential oils provided additional protection over chitosan-only films, highlighting the synergistic effect of essential oils in enhancing moisture retention and textural stability. These results suggest that maintaining higher water activity and stable texture through effective coatings can improve consumer acceptability and potentially extend shelf life. Statistical correlation analyses between water activity, dry matter, and textural parameters could further substantiate these relationships in future studies.

### 3.1.2. Measurement of the Color Characteristics of Cecil Cheese Samples

Color is one of the most critical sensory attributes shaping consumers' first impressions of food products and directly influences perceived quality [58]. When examining the L* (lightness-darkness) values*, high L* values were observed in all samples at the beginning, indicating that the cheeses had a light color. Notably, the industrial production + chitosan + thyme oil (EKE) group exhibited the highest L* value of 83.70 on day 1. In contrast, the L* value of the traditional production + chitosan + thyme oil (GKE) sample value decreased to 73.10 by day 30, suggesting darkening of the surface color during storage. Although slight fluctuations in L* values were observed in most chitosan-coated samples during storage, overall color retention was maintained. Regarding the a* (green-red axis*)

values, positive values indicate that all samples possessed reddish tones. The highest a* value was 8.01 on day 15 in the industrial production + chitosan + thyme oil (EKİ) sample. Over time, a decreasing trend in a* values was observed in all samples, reflecting a reduction in red tones during storage. In the traditional production + chitosan and rosemary oil (GBİ) group, the a* value decreased to 6.32 by day 45 (Table 4). It can be inferred that the addition of thyme oil has a positive effect on color stability in this context. The b* (blue-yellow axis)* values showed positive results across all samples, indicating a dominance of yellow tones in all groups. The highest b* value, 28.32, was determined in the industrial production + chitosan + thyme oil (EKE) sample on day 45. This result suggests that coatings containing thyme oil are effective in preserving yellow tones. The h (hue angle) parameter provides information about the overall color tone of the cheese. The h value reached its highest level of 76.34 on day 45 in the EKE group, while it dropped to its lowest value of 63.77 on day 15 for the GBİ sample. This difference highlights the impact of production technique and film composition on color tone.

Yüceer (2017) reported slight decreases over time in color values of Kashar cheese coated with chitosan films but noted better color stability compared to the control group [44]. Similarly, in this study, L*, a*, and b* values in chitosan-containing samples remained relatively stable. Karakuş (2021) observed that color values changed more rapidly in cheeses produced by traditional methods, whereas volatile oil additions preserved tone in industrially produced samples [45]. Our study also identified high L* and b* values in the oil combination of industrial production and thyme oil, consistent with the literature. Likewise, Özkan (2018) reported that the addition of thyme oil notably increased L* values and enhanced b* values, findings that parallel those in the EKE group [46]. However, some decreases in a* and h values observed in our study may be explained by the effects of volatile oils on pigment stability, as reported by Casalini and Giacinti (2023) [59]. That is, although film components may slow the oxidation of certain color components, a decline in reddish tones over time is possible.

In conclusion, chitosan-based edible film coatings, especially when combined with volatile oils, effectively preserved the color characteristics of Cecil cheese, with industrially produced samples benefiting more from this protection. Samples containing thyme oil (particularly EKE) maintained yellow tones, preserved high L* values, and demonstrated the best overall color stability.

The decreases in a* values observed in our study can be explained by the effects of essential oils on pigment stability reported by Vega et al. (2023) [60]. Although the film components slowed oxidation, a reduction in reddish tones over time was observed in their study. An increase in b* values was also noted. Özkan (2018) similarly reported that the use of thyme oil in coatings increased b* values and improved the preservation of yellow tones [46]. Yüceer (2017) stated that chitosan films maintained color stability despite minor decreases over time [44]. Karakuş (2021) found that color changes occurred more rapidly in traditionally produced cheeses, while coatings containing essential oils provided tone preservation, especially in industrial samples [45]. These findings are consistent with our results, where the EKE group exhibited the best preservation of L* and b* values. Artiga-Artigas et al. (2017) also reported that thyme oil in coatings preserved yellowness (b*) and increased brightness (L*) [21]. Martillanes et al. (2017) and Ribeiro-Santos et al. (2017) emphasized that phenolic compounds (e.g., carvacrol, thymol, and rosmarinic acid) in essential oils possess antioxidant properties that delay oxidative changes and help preserve the visual quality of cheese [17,61].

**Table 4.** Color analysis values of Cecil cheese samples.

| Parameter | Sample Code | Storage Days | | | |
|---|---|---|---|---|---|
| | | **Day 1** | **Day 15** | **Day 30** | **Day 45** |
| L* | G K | 81.09 BCa ± 1.32 | 72.03 Ab ± 1.36 | 80.60 Aa ± 1.19 | 79.24 Aa ± 0.77 |
| | GKİ | 80.69 Ca ± 1.12 | 74.00 Ab ± 1.42 | 70.71 Dc ± 2.07 | 78.36 ABa ± 1.36 |
| | GBİ | 82.07 BCa ± 1.11 | 79.32 Ab ± 0.22 | 72.36 CDc ± 1.02 | 73.36 Cc ± 1.02 |
| | GKE | 81.43 BCa ± 0.69 | 79.80 Aa ± 0.06 | 73.10 Cb ± 1.45 | 74.10 Cb ± 0.86 |
| | EK | 79.01 Da ± 0.98 | 76.24 Aa ± 0.91 | 80.45 Aa ± 0.16 | 79.38 Aa ± 1.39 |
| | EKİ | 82.43 ABa ± 0.17 | 76.06 Ac ± 1.23 | 79.55 ABb ± 0.76 | 73.72 Cd ± 0.39 |
| | EBİ | 82.22 ABCa ± 0.24 | 74.73 Ab ± 1.62 | 78.54 ABab ± 0.52 | 69.12 Dc ± 0.78 |
| | EKE | 83.70 Aa ± 0.05 | 80.52 Ab ± 1.47 | 77.87 Bc ± 0.66 | 77.19 Bc ± 0.49 |
| a* | G K | 7.21 BCDa ± 0.71 | 6.68 Aa ± 1.61 | 7.16 ABCa ± 0.64 | 6.71 BCDa ± 0.16 |
| | GKİ | 7.73 ABa ± 0.43 | 7.45 Aa ± 0.24 | 7.93 ABa ± 0.26 | 6.57 CDb ± 0.37 |
| | GBİ | 8.10 Aa ± 0.60 | 7.02 Ab ± 0.10 | 6.79 Cb ± 0.74 | 6.32 Db ± 0.24 |
| | GKE | 7.37 ABa ± 0.38 | 6.72 Aa ± 0.09 | 7.03 BCa ± 0.70 | 7.54 ABa ± 0.34 |
| | EK | 6.53 Da ± 0.10 | 6.98 Aa ± 0.09 | 7.62 ABCa ± 0.23 | 7.27 ABCa ± 0.53 |
| | EKİ | 7.50 ABa ± 0.09 | 8.01 Aa ± 0.14 | 8.10 Aa ± 0.36 | 7.78 Aa ± 0.93 |
| | EBİ | 6.60 CDbc ± 0.06 | 7.00 Aab ± 0.37 | 7.57 ABCa ± 0.49 | 6.29 Dc ± 0.05 |
| | EKE | 7.27 BCa ± 0.09 | 7.05 Aa ± 0.42 | 6.95 Ca ± 0.02 | 6.77 BCDa ± 0.51 |
| b* | G K | 26.48 ABa ± 0.59 | 23.74 Ab ± 2.02 | 26.50 Aa ± 0.24 | 26.61 ABa ± 0.58 |
| | GKİ | 26.12 ABa ± 0.55 | 26.25 Aa ± 0.61 | 25.36 Aa ± 1.18 | 26.12 Ba ± 0.94 |
| | GBİ | 26.85 Aa ± 0.13 | 25.78 Ab ± 0.06 | 26.08 Ab ± 0.50 | 26.08 Bb ± 0.45 |
| | GKE | 25.90 Bab ± 0.36 | 24.69 Ab ± 0.14 | 25.84 Aab ± 1.87 | 27.27 ABa ± 1.24 |
| | EK | 26.12 ABa ± 0.58 | 27.99 Aa ± 0.45 | 27.09 Aa ± 0.71 | 28.32 Aa ± 2.45 |
| | EKİ | 26.87 Aab ± 0.11 | 24.78 Ab ± 1.20 | 25.80 Ab ± 1.63 | 28.33 Aa ± 0.95 |
| | EBİ | 26.48 ABa ± 0.62 | 24.88 Ab ± 0.31 | 26.66 Aa ± 0.70 | 27.18 ABa ± 0.08 |
| | EKE | 26.53 ABb ± 0.10 | 25.74 Ac ± 0.36 | 27.31 Aa ± 0.21 | 27.47 ABa ± 0.13 |
| Δη | G K | 74.77 ABa ± 1.27 | 63.53 Db ± 1.18 | 74.88 ABa ± 1.17 | 75.84 ABCa ± 0.41 |
| | GKİ | 73.51 BCa ± 0.71 | 65.44 Cb ± 0.63 | 72.63 Ca ± 0.67 | 74.10 Ca ± 1.82 |
| | GBİ | 73.22 Ca ± 1.13 | 63.77 Db ± 0.22 | 75.23 Aa ± 1.31 | 75.23 ABCa ± 1.31 |
| | GKE | 74.12 BCa ± 0.62 | 64.12 Db ± 0.62 | 74.78 ABa ± 0.92 | 74.54 BCa ± 0.17 |
| | EK | 75.95 Aa ± 0.52 | 76.21 Aa ± 0.11 | 73.77 ABCb ± 0.14 | 75.57 ABCa ± 0.47 |
| | EKİ | 74.40 BCab ± 0.13 | 74.23 Bab ± 0.15 | 73.00 BCb ± 1.51 | 75.65 ABCa ± 0.43 |
| | EBİ | 75.99 Aa ± 0.22 | 75.93 Aa ± 0.21 | 74.16 ABCb ± 0.83 | 76.93 Aa ± 0.86 |
| | EKE | 74.66 ABab ± 0.14 | 74.55 Bb ± 0.14 | 74.97 ABab ± 1.44 | 76.34 ABa ± 0.89 |

*: mean ± standard deviation. **Note**: Values are expressed as **mean ± standard deviation**. **A–D**: Capital letters in the same column indicate statistically significant differences between sample types on the same storage day (*p* < 0.05). **a–d**: Lowercase letters in the same row indicate statistically significant differences within the same sample over different storage times (*p* < 0.05). **Sample Codes**: **GK**: Traditional production, control group, **GKİ**: Traditional production, Cecil cheese coated with chitosan, **GBİ**: Traditional production, Cecil cheese coated with chitosan + rosemary essential oil, **GKE**: Traditional production, Cecil cheese coated with chitosan + thyme essential oil, **EK**: Industrial production, control group, **EKİ**: Industrial production, Cecil cheese coated with chitosan, **EBİ**: Industrial production, Cecil cheese coated with chitosan + rosemary essential oil, **EKE**: Industrial production, Cecil cheese coated with chitosan + thyme essential oil.

Yangılar (2015) and Iqbal et al. (2021) demonstrated the effectiveness of chitosan and whey protein-based coatings in preserving cheese color [51,62]. Moreover, plant extracts have been observed to cause slight color changes due to their natural pigments; however, these changes are generally considered acceptable in sensory evaluations [63].

In particular, chitosan-based edible coatings enriched with essential oils effectively preserved the color characteristics of Cecil cheese. Industrially produced samples benefited more from this protective effect. Among all treatments, samples containing thyme oil (especially EKE) best preserved yellow tones, maintained high L* values, and provided the greatest overall color stability.

3.1.3. Evaluation of the Microbiological Data of Cecil Cheese Samples

According to the results obtained on the first day, the traditional production control group (GK) exhibited the highest total aerobic mesophilic bacteria (TAMB) (4.25 log CFU/g), while the industrial production group coated with chitosan + thyme oil (EKE) showed the lowest value (3.05 log CFU/g). This indicates that the combination of essential oils used as coating materials had a positive effect on the initial microbial load. In the later stages of storage, a general decreasing trend was observed in all samples. Particularly by day 30, the TAMB count in the EKE group decreased to 2.79 log CFU/g—the lowest value recorded in this study. This suggests that the synergy of chitosan and thyme oil in the coating effectively inhibited microbial growth. Similarly, in the EBİ group (chitosan + rosemary oil), the TAMB count on day 30 was also notably low (2.86 log CFU/g). In the traditional production groups, the coating applications also significantly reduced the microbial load compared to the control group. For instance, on day 15, while the GK sample had a TAMB value of 3.32 log CFU/g, the GKE sample was measured at 3.09 log CFU/g. This difference indicates that the antimicrobial effects of essential oils manifested effectively on the cheese surface. By the end of the storage period (day 45), the control groups showed a tendency toward microbial increase (e.g., GK: 3.43 log CFU/g), whereas the coated samples largely limited this increase. Specifically, the EBI (3.13 log CFU/g) and EKE (3.21 log CFU/g) groups maintained TAMB levels below those of the control groups (Table 5). These findings support the potential for chitosan- and essential oil-based edible film applications in slowing microbial growth throughout the shelf life.

The TAMB values obtained in this study are consistent with results from similar applications reported in the literature. For example, Karakuş (2021) reported that chitosan-based films applied to Kashar cheese maintained TAMB counts around 3.30 log CFU/g by day 30 [45]. In our study, some groups exhibited even lower values, revealing the synergistic effect of essential oil supplementation. Yüceer (2017) observed that in uncoated traditional Kashar cheese groups, TAMB values reached up to 4.50 log CFU/g by day 45, while chitosan-coated groups maintained values around 3.10 log CFU/g, confirming the effect of chitosan [44]. International studies such as those by Arfat et al. (2015) and Casalini and Giacinti (2023) have frequently reported the antimicrobial potential of chitosan and essential oil combinations, particularly highlighting the effectiveness of phenolic compound-rich oils such as thyme and rosemary [54,59]. TAMB results obtained in our study demonstrate that chitosan-based films enriched with essential oils are effective in maintaining microbial stability in high-moisture, spoilage-prone dairy products such as Cecil cheese. This contributes significantly to both consumer safety and shelf-life extension.

In the examined Cecil cheese samples, total yeast and mold counts varied depending on production method, type of coating applied, and storage duration. While yeast and mold levels were initially low in all samples, an increasing trend was observed as storage progressed. However, this increase was more limited in samples treated with edible films compared to the control groups. In the traditional production group, the control sample (GK) exhibited the highest yeast-mold level on day 15 (3.41 log CFU/g). A significant decline was noted on day 30 (2.21 log CFU/g), followed by a resurgence on day 45 (2.85 log CFU/g). In contrast, coated samples in the traditional group generally

followed a more stable trend. In particular, the sample coated with chitosan film enriched with thyme oil (GKE) showed low microbial loads on both day 15 and day 30 (Table 5).

**Table 5.** Results of microbiological analyses performed on Cecil cheese samples.

| Parameter | Sample Code | Storage Days | | | |
|---|---|---|---|---|---|
| | | **Day 1** | **Day 15** | **Day 30** | **Day 45** |
| Total mesophilic aerobic bacteria (log CFU/g) | G K | 4.25 Aa ± 0.02 | 3.32 Ac ± 0.25 | 3.22 Ad ± 0.02 | 3.43 Ab ± 0.05 |
| | GKİ | 4.11 Ca ± 0.01 | 3.22 Bc ± 0.25 | 3.20 Ac ± 0.15 | 3.30 Bb ± 0.01 |
| | GBİ | 4.05 Da ± 0.01 | 3.11 Dc ± 0.15 | 3.11 Cc ± 0.01 | 3.28 Bb ± 0.01 |
| | GKE | 4.06 Da ± 0.03 | 3.09 Dd ± 0.05 | 3.18 Ac ± 0.15 | 3.26 Cb ± 0.01 |
| | EK | 3.14 Ec ± 0.02 | 3.09 Dd ± 0.01 | 3.16 Bb ± 0.01 | 3.24 Ca ± 0.57 |
| | EKİ | 4.20 Ba ± 0.01 | 3.20 BCb ± 0.11 | 2.94 Dc ± 0.01 | 3.21 Db ± 0.15 |
| | EBİ | 3.07 Fc ± 0.01 | 3.20 BCa ± 0.05 | 2.86 Ed ± 0.05 | 3.13 Eb ± 0.15 |
| | EKE | 3.05 Fc ± 0.03 | 3.19 Cb ± 0.05 | 2.79 Fd ± 0.01 | 3.21 Da ± 0.57 |
| Total yeast and mold count (log CFU/g) | G K | 3.12 Ab ± 0.05 | 3.41 Aa ± 0.15 | 2.21 Fd ± 0.01 | 2.85 Ac ± 0.57 |
| | GKİ | 2.84 Cc ± 0.15 | 3.11 Da ± 0.01 | 2.91 Db ± 0.02 | 2.73 Cd ± 0.11 |
| | GBİ | 3.00 Ba ± 0.57 | 2.82 Fb ± 0.34 | 2.81 Eb ± 0.02 | 2.62 Dc ± 0.04 |
| | GKE | 2.82 Db ± 0.02 | 2.71 Gc ± 0.15 | 2.90 Da ± 0.01 | 2.73 Cc ± 0.57 |
| | EK | 2.55 Ed ± 0.02 | 3.31 Ba ± 0.15 | 3.06 Bb ± 0.05 | 2.71 Cc ± 0.17 |
| | EKİ | 1.20 Hc ± 0.01 | 3.23 Ca ± 0.02 | 3.25 Aa ± 0.05 | 2.71 Cb ± 0.57 |
| | EBİ | 2.05 Fc ± 0.01 | 2.92 Ea ± 0.02 | 2.94 Ca ± 0.01 | 2.70 Cb ± 0.11 |
| | EKE | 1.62 Gd ± 0.02 | 2.70 Gc ± 0.20 | 2.90 Da ± 0.03 | 2.81 Bb ± 0.17 |
| Total coliform count (log CFU/g) | G K | <1 | <1 | <1 | <1 |
| | GKİ | <1 | <1 | <1 | <1 |
| | GBİ | <1 | <1 | <1 | <1 |
| | GKE | <1 | <1 | <1 | <1 |
| | EK | <1 | <1 | <1 | <1 |
| | EKİ | <1 | <1 | <1 | <1 |
| | EBİ | <1 | <1 | <1 | <1 |
| | EKE | <1 | <1 | <1 | <1 |
| Total lactic acid bacteria count (log CFU/g) | G K | 4.13 Ba ± 0.15 | 2.74 Fd ± 0.05 | 3.00 Cc ± 0.57 | 3.16 DEb ± 0.57 |
| | GKİ | 3.26 Ca ± 0.01 | 3.11 Db ± 0.05 | 3.01 Cc ± 0.02 | 3.25 Ba ± 0.05 |
| | GBİ | 4.16 Aa ± 0.57 | 2.86 Ed ± 0.01 | 3.01 Cc ± 0.01 | 3.20 Cb ± 0.05 |
| | GKE | 4.16 Aa ± 0.57 | 2.58 Gd ± 0.11 | 3.02 Cc ± 0.02 | 3.16 Eb ± 0.07 |
| | EK | 2.74 Ed ± 0.57 | 3.26 Ca ± 0.05 | 3.00 Cc ± 0.05 | 3.17 Db ± 0.17 |
| | EKİ | 2.62 Gb ± 0.05 | 3.56 Aa ± 0.01 | 3.56 Aa ± 0.01 | 3.55 Aa ± 0.05 |
| | EBİ | 2.77 Db ± 0.51 | 3.54 Ba ± 0.01 | 3.56 Aa ± 0.11 | 3.56 Aa ± 0.50 |
| | EKE | 2.72 Fd ± 0.57 | 3.27 Cc ± 0.05 | 3.43 Bb ± 0.57 | 3.56 Aa ± 0.18 |

Mean ± standard deviation. **Note**: Values are expressed as **mean ± standard deviation**. **A–H**: Capital letters in the same column indicate statistically significant differences between sample types on the same storage day (***p* < 0.05**). **a–d**: Lowercase letters in the same row indicate statistically significant differences within the same sample over different storage times (***p* < 0.05**). **Sample Codes**: **GK**: Traditional production, control group, **GKİ**: Traditional production, Cecil cheese coated with chitosan, **GBİ**: Traditional production, Cecil cheese coated with chitosan + rosemary essential oil, **GKE**: Traditional production, Cecil cheese coated with chitosan + thyme essential oil, **EK**: Industrial production, control group, **EKİ**: Industrial production, Cecil cheese coated with chitosan, **EBİ**: Industrial production, Cecil cheese coated with chitosan + rosemary essential oil, **EKE**: Industrial production, Cecil cheese coated with chitosan + thyme essential oil.

In the industrial production group, the control sample (EK) had the highest value on day 15 (3.31 log CFU/g), which then decreased to 2.71 log CFU/g by day 45. This suggests that microbial growth can be controlled over time under industrial conditions, although

coating applications more effectively limited microbial activity. The industrial production sample coated with chitosan (EKİ) had a particularly low yeast-mold level on the first day (1.20 log CFU/g), which rose to 3.23 log CFU/g on day 15, then declined again to 2.71 log CFU/g by day 45. In the industrial group coated with chitosan + rosemary oil (EBİ), microbial growth remained moderate, and stable values were maintained on days 30 and 45. These results demonstrate that both the method of production and the composition of the applied films are critical in determining yeast and mold growth. Particularly, chitosan-based films combined with essential oils exhibited stronger antimicrobial effects, helping to limit microbial load. In this context, coatings with chitosan + thyme oil were more effective in the traditional production group, while chitosan + rosemary oil coatings showed better results in the industrial group.

All microbiological data (TAMB, yeast-mold, and lactic acid bacteria) were analyzed using one-way ANOVA followed by Tukey's multiple comparison tests in SPSS 25.0. The analyses revealed that coatings containing chitosan and essential oils significantly reduced total aerobic mesophilic bacteria, yeast-mold, and lactic acid bacteria counts compared to control groups ($p < 0.05$). In particular, chitosan and thyme oil coatings exhibited the most pronounced reductions in TAMB and yeast-mold counts in both traditional and industrial samples. Lactic acid bacteria levels in some industrial samples increased slightly with essential oil-enriched coatings, indicating selective antimicrobial activity and potential support for beneficial microflora at low essential oil concentrations. These findings confirm that the film composition has a functionally and statistically reliable effect on microbial control.

Similarly, Özkan (2018) reported significantly reduced yeast-mold growth in Kashar cheese coated with edible films compared to control samples [46]. Karakuş (2021) also emphasized that chitosan-based coatings suppressed both yeast-mold and total mesophilic bacterial counts, positively influencing the microbiological quality of cheeses [45]. Yüceer (2017) stated that chitosan films enriched with essential oils extended shelf life due to their natural antimicrobial properties [44]. The findings of this study are consistent with those in the literature and support that chitosan-based active film applications offer a promising alternative for microbial control in sensitive dairy products such as Cecil cheese.

Coliform bacteria are critical indicator microorganisms for food safety and are frequently used to detect possible hygiene issues during production and storage. In this study, analyses were conducted to assess the effect of different film applications on coliform bacteria growth in Cecil cheese samples. In all samples, values remained below the detectable level (<1 log CFU/g) throughout the storage period (Table 5). In both traditional and industrial production cheese samples—whether control, chitosan-coated, or with chitosan + rosemary oil or chitosan + thyme oil—no coliform bacteria were detected during the four storage intervals. This indicates that hygiene standards during production were sufficiently maintained and that the applied coatings provided a protective barrier against the growth of this bacterial group. The findings align with those reported in similar studies. For instance, Arfat et al. (2015) highlighted the inhibitory effect of chitosan-based coatings on coliform bacteria in Kashar cheese [54]. Similarly, Karakuş (2021) reported that coliform growth was significantly limited in Cecil cheeses treated with edible films compared to control samples [45]. These data suggest that the film compositions not only function as physical barriers but also play an active role in preventing microbial contamination. Furthermore, the synergistic effect of chitosan and essential oils likely contributes to the inhibition of microbial proliferation.

Lactic acid bacteria are particularly important in dairy products as part of the natural microbiota and for ensuring proper fermentation processes. In this study, changes in the population of lactic acid bacteria in traditional and industrially produced Cecil cheese samples with various edible film coatings were monitored over four storage periods (days 1, 15, 30, and 45). On the first day of storage, the highest levels of lactic acid bacteria were observed in traditionally produced samples coated with chitosan and rosemary oil (GBİ and GKE: 4.16 log CFU/g), while the lowest values were found in industrial production samples coated with chitosan and thyme oil (EKE: 2.72 log CFU/g) and chitosan only (EKİ: 2.62 log CFU/g) (Table 5). This indicates that the initial microbial populations vary based on production method and film composition. Throughout the storage period, a general decreasing trend was noted, but from day 15 onwards, a significant increase in lactic acid bacteria was observed in industrial production groups, particularly in samples EKİ, EBİ, and EKE. Notably, in samples coated with chitosan + rosemary oil (EBİ) and chitosan + thyme oil (EKE), lactic acid bacteria levels increased to approximately 3.56 log CFU/g by day 30 and day 45—a statistically significant rise ($p < 0.05$) at each point. This may be attributed to the ability of phenolic compounds in essential oils to support probiotic bacterial growth at low concentrations. In the traditional production groups, lactic acid bacteria levels remained more stable throughout storage. Following the high initial counts, a slight decrease and subsequent stabilization were observed. For example, the traditional control group (GK) had one of the highest initial values on day 1 (4.13 log CFU/g), dropped to 2.74 log CFU/g by day 15, and then increased again to reach 3.00–3.16 log CFU/g by days 30 and 45. The lactic acid bacteria counts in this study align with findings reported by Karakuş (2021), who observed values between 2.80 and 4.00 log CFU/g in chitosan-coated Kashar cheese [43]. Similarly, Yüceer (2017) found lactic acid bacteria values between 3.10 and 3.80 log CFU/g in traditional white cheese and noted that coating applications could exert a suppressive or balancing effect on these bacteria [44]. Interestingly, the lactic acid bacteria counts on the 45th day, observed in industrially produced samples in our study (e.g., EBİ: 3.56 log CFU/g), were higher than those reported in the literature. This may be due to the dual role of essential oil components: their antimicrobial effect and their capacity, at appropriate concentrations, to support probiotic bacterial growth without suppression [40,64]. Both the production method (traditional vs. industrial) and the composition of the edible films directly influenced lactic acid bacteria development. Industrially produced samples coated with essential oil-enriched films demonstrated more stable lactic acid bacteria profiles, highlighting their potential as functional preservation systems.

In this study, the effects of edible films containing chitosan, chitosan + thyme oil, and chitosan + rosemary oil on the microbiological quality of Cecil cheese were monitored over a 45-day storage period. The results obtained are consistent with previously published data. Dikbaş et al. (2010) reported total mesophilic aerobic bacteria (TMAB), yeast-mold counts, lactic acid bacteria, and coliform counts in Cecil cheese samples collected from the Erzurum region as $1.6 \times 10^6$, $7 \times 10^4$, $1.8 \times 10^5$, and <10 CFU/g, respectively [49]. Similarly, Sengül et al. (2009) calculated these counts as 6.65, 3.54, 5.57, and 0.87 log CFU/g, respectively [35]. Compared to our findings, Sengül et al. (2009) reported higher microbial loads in their samples [35]. These differences may result from the raw materials used, production processes, additives, and storage conditions. Moreover, the edible films and coatings used in this study contributed to the microbiological safety of the product. Lower total mesophilic aerobic bacteria (TMAB) and yeast-mold counts were recorded in the coated samples, indicating enhanced microbial safety. The absence of coliform bacteria also suggests that the products were produced and stored under hygienic conditions.

TMAB counts were significantly lower in samples coated with chitosan and thyme oil, particularly in industrially produced cheese. This finding highlights the strong antimi-

crobial effect of chitosan-based films, contributing to extended shelf life [44,45,64,65]. The enhanced antimicrobial activity of thyme oil can be attributed to its phenolic compounds, such as carvacrol and thymol, which disrupt microbial membranes [40,55,61]. Yeast and mold growth was significantly inhibited in samples coated with essential oil-enriched films. These results are in line with previous studies reporting the antifungal effectiveness of thyme and rosemary oils [46,47,66–68]. For instance, Kavas and Kesenkas (2018) and Yüceer (2017) reported delayed mold growth and extended shelf life in cheese coated with thyme-oil-enriched chitosan films [44,65]. Coliform bacteria were below detectable levels (<1 log CFU/g) in all groups, which may be attributed to hygienic production and the protective nature of the coatings. These findings are consistent with the results of Bleoancă et al. (2020) and Karakuş (2021), who reported coliform inhibition through chitosan and essential oil films [45,55]. Lactic acid bacteria (LAB) counts were higher in the coated industrial samples, suggesting that the coatings did not inhibit beneficial microflora. Low concentrations of essential oils may have supported LAB growth. This observation aligns with previous studies [4,19,40]. It suggests that active coatings can selectively inhibit spoilage microorganisms while preserving or even promoting probiotic populations. The microbiological findings of this study confirm that chitosan and essential-oil-enriched edible films are effective in enhancing microbial stability and extending the shelf life of high-moisture dairy products such as Cecil cheese [44,45,53].

3.1.4. Evaluation of the Sensory Analysis of Cecil Cheese Samples

Overall, color scores remained high across all groups. The highest scores were observed in GBİ, GKE, GKİ and GK samples, with scores ranging between 8.66 and 8.78. The industrial control group (EK) and its coated variants also received similarly high scores, although traditional groups such as GKE were found to be preferable in terms of color. No statistically significant change was observed during storage ($p > 0.05$), indicating that films containing chitosan and essential oils contributed to color stability. The highest texture scores (8.56) were recorded in GKE samples on both day 30 and day 45. GK and GKİ samples also exhibited similarly high scores. This suggests that chitosan-based coatings limited moisture loss from the surface of the product, resulting in a more pleasant texture. In contrast, lower scores were noted in EK, EKE and EBİ samples. The textural structure of traditionally produced cheeses was more positively evaluated by the panelists. In terms of odor, traditional samples coated with chitosan and rosemary oil (GBI) and thyme oil (GKE) received relatively lower scores (around 6.00). This may be due to the intense and characteristic aroma profiles of essential oils, which were not well-received by some panelists. EKİ and EK samples received relatively higher odor scores, suggesting that industrial production contributed to a more standard aroma profile. GK and GKİ samples received taste scores as high as 8.00–8.72, and they were statistically more appreciated than other groups ($p < 0.05$). The traditional control sample especially stood out due to its characteristic flavor derived from natural production methods. In essential oil-containing groups—particularly EBİ—taste scores dropped to as low as 4.66. This indicates that the bitter or sharp aromatic notes of rosemary oil may not have fully harmonized with the flavor profile of Cecil cheese. Consumer acceptance may be enhanced by reducing the amount of essential oils used and by developing new combinations. In saltiness evaluations, GKİ and GK samples were more favorable, while EBİ samples received lower scores. This suggests that edible films may influence salt release and that the brining conditions used in traditional production are more effective. GK and GKİ samples consistently received high overall acceptability scores (between 8.33 and 8.73). These groups maintained consumer preference throughout storage due to the benefits of traditional production and the protective effect of chitosan coatings. However, EBİ received low scores, indicating that the intense rosemary aroma negatively

affected overall acceptance. The sensory analysis data are presented in Figure 6. Based on these graphs, the scores obtained by the samples can be more easily distinguished according to the storage periods. EKE and EKİ groups showed moderate scores, ranging between 7.00 and 7.33. According to the sensory analysis results, the traditional control (GK) and traditional chitosan-coated (GKİ) samples received the highest purchase intent score with an average of 1.00, which corresponds to the "definitely would buy" option. This reflects a strong preference among consumers for products that possess a familiar cheese profile. In contrast, EBİ and EKE samples had an average purchase intent score of 3.00, which corresponds to the "definitely would not buy" category (Table 6). Sensory outcomes corroborate the importance of optimizing essential oil concentration in chitosan films. While antimicrobial activity increases with higher essential oil levels, excessive concentrations—particularly rosemary oil—can negatively affect taste and aroma. The literature supports that controlled polymer-essential oil interactions enable sustained antimicrobial efficacy while minimizing sensory compromise [40,42,54]. In this study, thyme oil at optimized levels achieved this balance, maintaining both microbial safety and consumer acceptance. These results emphasize that edible film design must consider both functional protection and sensory harmony to ensure market success. This suggests that volatile oils used in industrial production may unexpectedly influence the sensory characteristics of cheese and leave a negative impression on some consumers. These results indicate that consumer preference is shaped not only by sensory quality but also by the familiarity and traditional characteristics of the product. Similarly, studies by Burt (2004) and Fadlıoğlu and Ertan (2013) have emphasized that consumer habits significantly influence the acceptance of products produced using new technologies or containing additives [40,69]. Therefore, optimizing newly developed edible film applications to match traditional flavor profiles as closely as possible is crucial for market success.

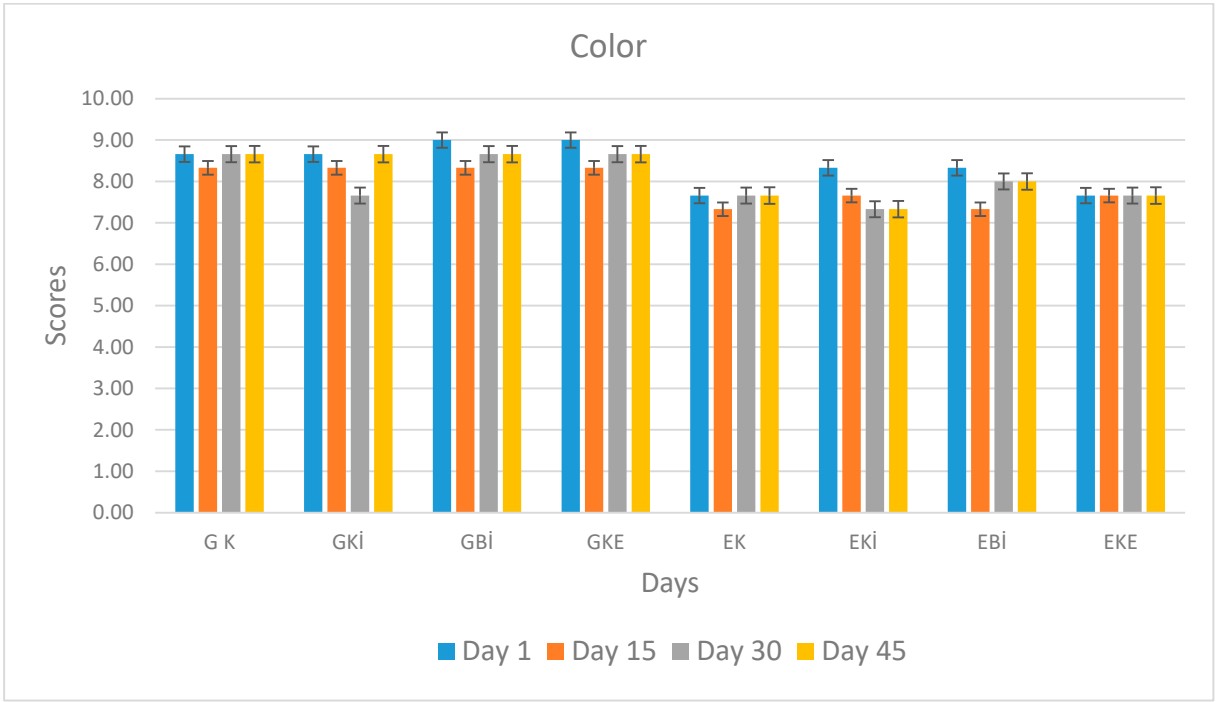

**Figure 6.** *Cont*.

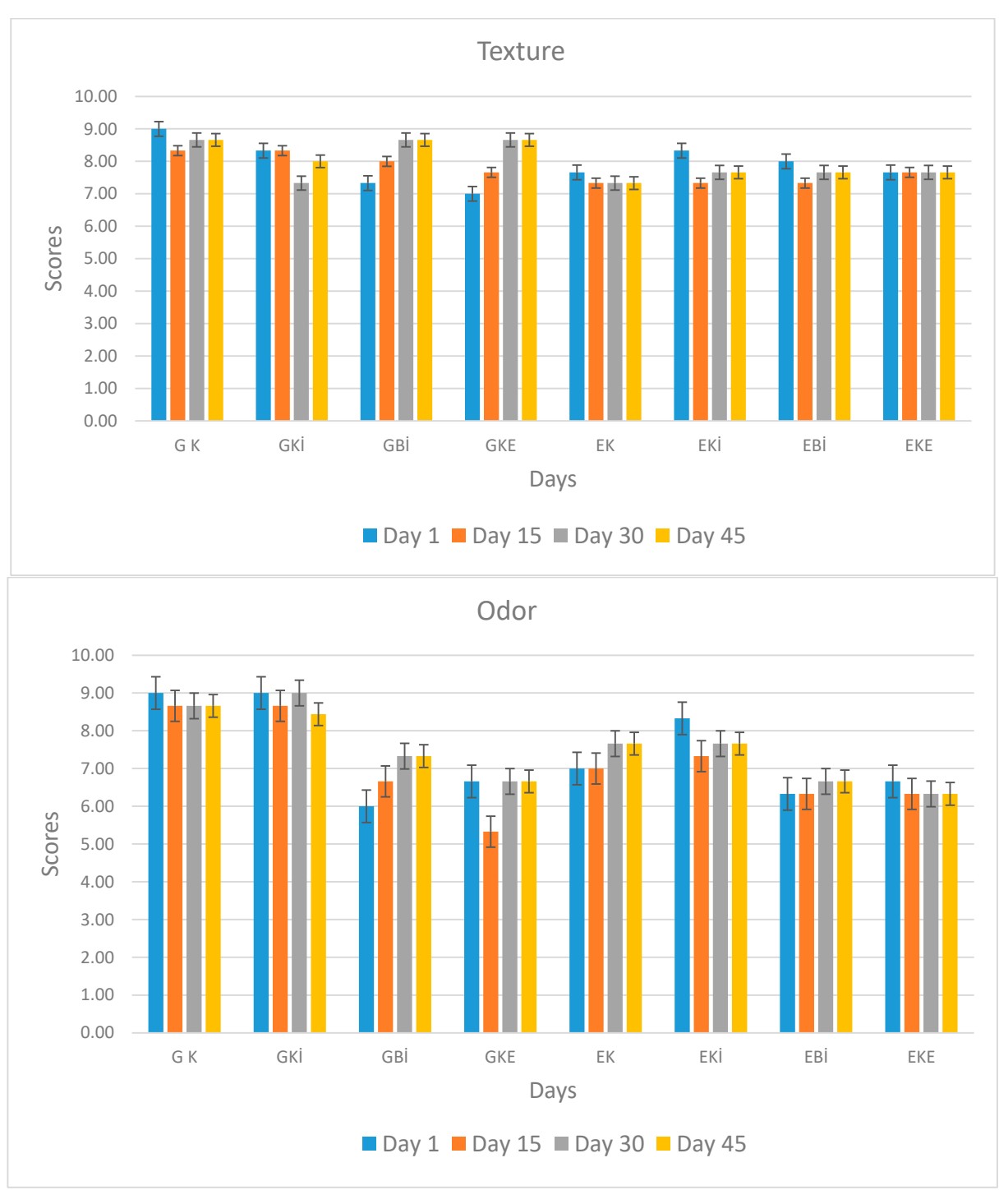

**Figure 6.** *Cont.*

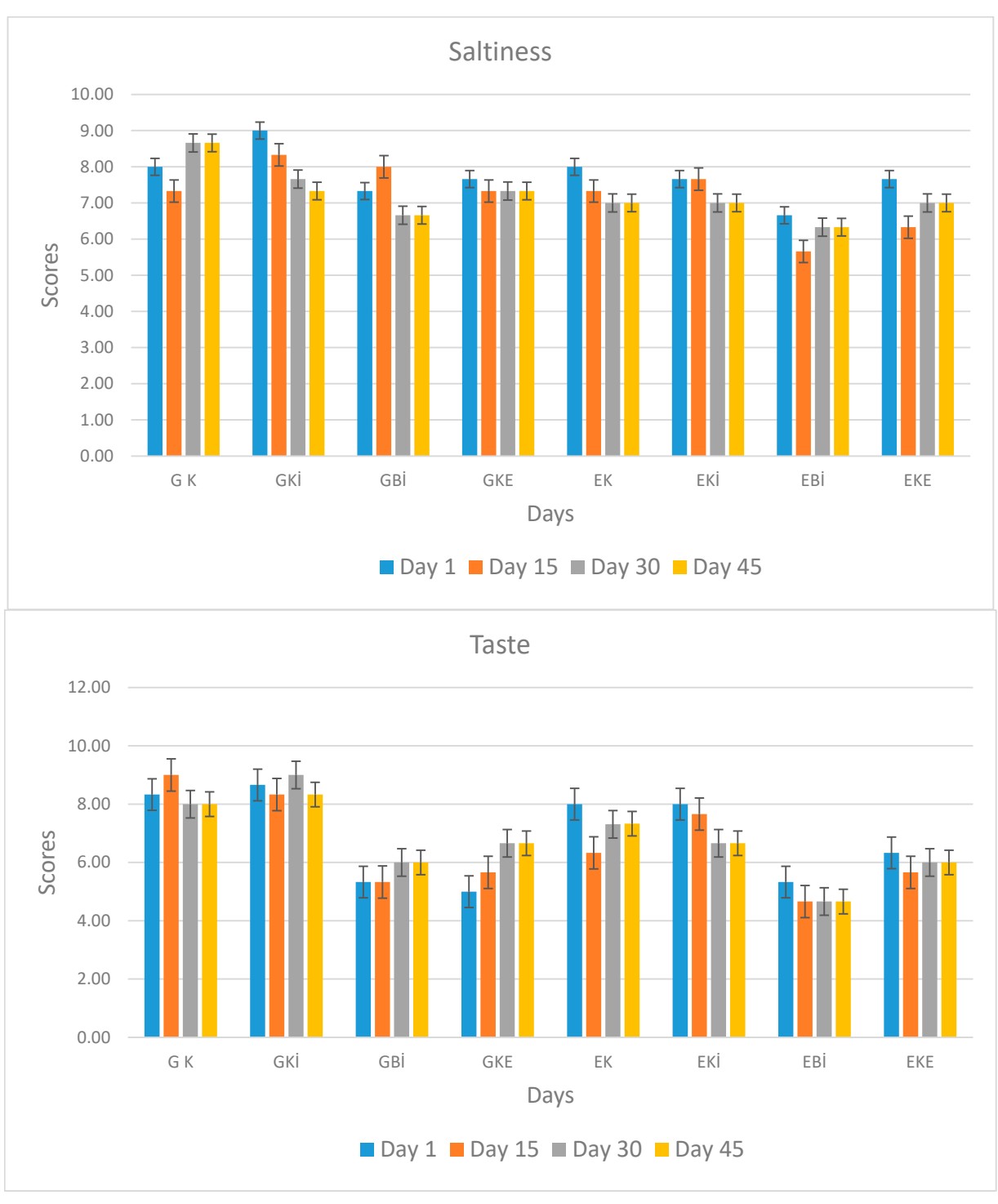

**Figure 6.** *Cont.*

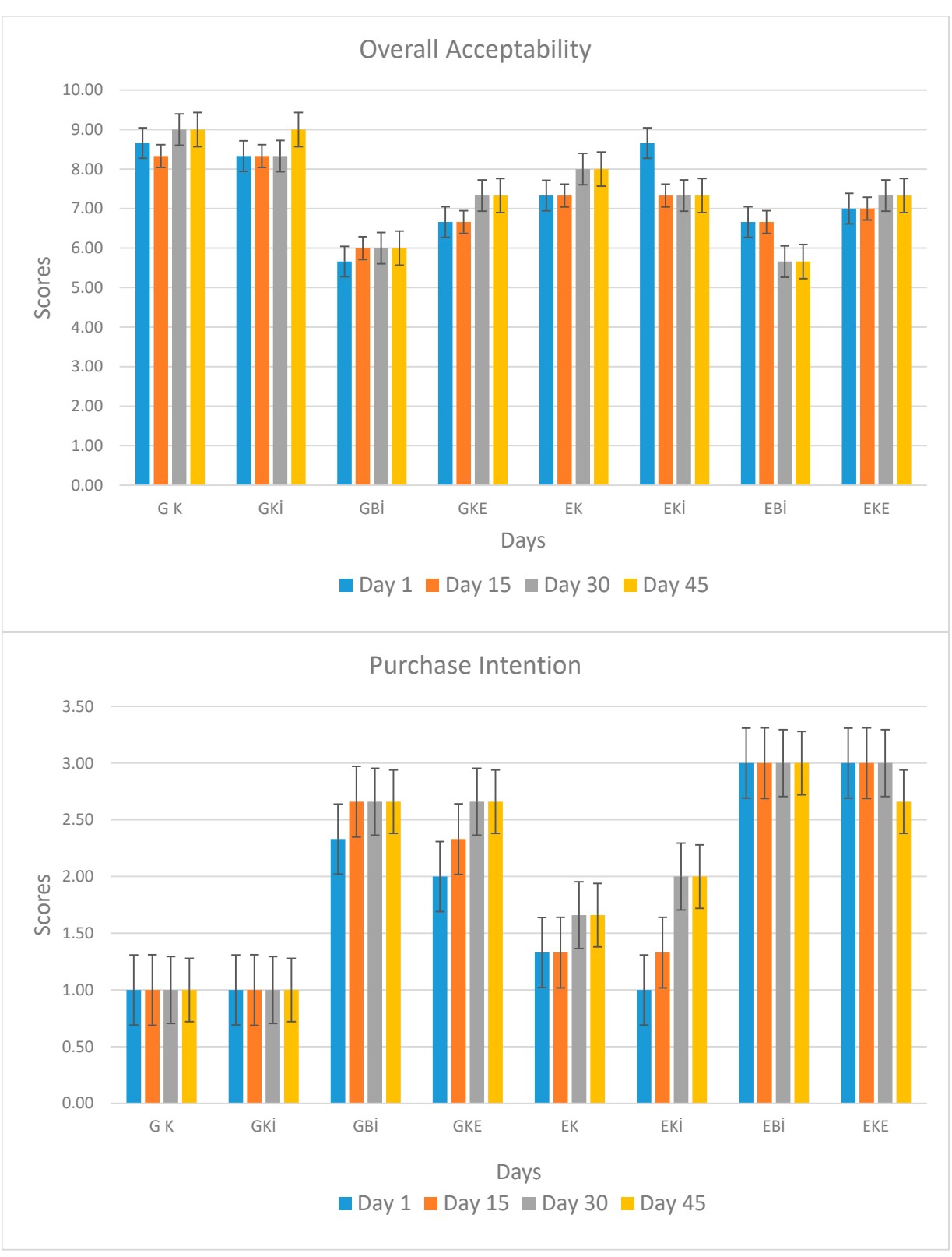

**Figure 6.** Graphs of sensory evaluation results. **Sample Codes**: **GK**: Traditional production, control group, **GKİ**: Traditional production, Cecil cheese coated with chitosan, **GBİ**: Traditional production, Cecil cheese coated with chitosan + rosemary essential oil, **GKE**: Traditional production, Cecil cheese coated with chitosan + thyme essential oil, **EK**: Industrial production, control group, **EKİ**: Industrial production, Cecil cheese coated with chitosan, **EBİ**: Industrial production, Cecil cheese coated with chitosan + rosemary essential oil, **EKE**: Industrial production, Cecil cheese coated with chitosan + thyme essential oil.

**Table 6.** Sensory analysis results of Cecil cheese samples.

| Parameter | Sample Code | Storage Days | | | |
|---|---|---|---|---|---|
| | | Day 1 | Day 15 | Day 30 | Day 45 |
| Color | G K | 8.66 $^{Aa}$ ± 0.27 | 8.33 $^{Aa}$ ± 0.57 | 8.66 $^{Aa}$ ± 0.15 | 8.66 $^{Aa}$ ± 0.05 |
| | GKİ | 8.66 $^{Aa}$ ± 0.27 | 8.33 $^{Aa}$ ± 0.57 | 7.66 $^{ABa}$ ± 0.57 | 8.66 $^{Aa}$ ± 0.19 |
| | GBİ | 8.78 $^{Aa}$ ± 0.03 | 8.33 $^{Aa}$ ± 0.57 | 8.66 $^{Aa}$ ± 0.12 | 8.66 $^{Aa}$ ± 0.21 |
| | GKE | 8.15 $^{Aa}$ ± 0.05 | 8.33 $^{Aa}$ ± 0.57 | 8.66 $^{Aa}$ ± 0.24 | 8.66 $^{Aa}$ ± 0.06 |
| | EK | 7.66 $^{Ba}$ ± 0.57 | 7.33 $^{Aa}$ ± 0.57 | 7.66 $^{ABa}$ ± 0.57 | 7.66 $^{ABa}$ ± 0.57 |
| | EKİ | 8.33 $^{ABa}$ ± 0.57 | 7.66 $^{Aa}$ ± 0.57 | 7.33 $^{Ba}$ ± 0.57 | 7.33 $^{Ba}$ ± 0.57 |
| | EBİ | 8.33 $^{ABa}$ ± 0.57 | 7.33 $^{Ab}$ ± 0.57 | 8.00 $^{ABab}$ ± 0.50 | 8.00 $^{ABab}$ ± 0.50 |
| | EKE | 7.66 $^{Ba}$ ± 0.57 | 7.66 $^{Aa}$ ± 0.57 | 7.66 $^{ABa}$ ± 0.57 | 7.66 $^{ABa}$ ± 0.57 |
| Texture | G K | 8.52 $^{Aa}$ ± 0.05 | 8.33 $^{Aa}$ ± 0.18 | 8.48 $^{Aa}$ ± 0.27 | 8.30 $^{Aa}$ ± 0.57 |
| | GKİ | 8.23 $^{ABa}$ ± 0.37 | 8.40 $^{Aa}$ ± 0.15 | 7.33 $^{Ba}$ ± 0.57 | 8.00 $^{ABa}$ ± 0.50 |
| | GBİ | 7.33 $^{CDb}$ ± 0.57 | 8.00 $^{Aab}$ ± 0.35 | 8.36 $^{Aa}$ ± 0.27 | 8.36 $^{Aa}$ ± 0.25 |
| | GKE | 7.00 $^{Db}$ ± 0.15 | 7.66 $^{Ab}$ ± 0.57 | 8.56 $^{Aa}$ ± 0.37 | 8.46 $^{Aa}$ ± 0.07 |
| | EK | 7.66 $^{BCDa}$ ± 0.57 | 7.33 $^{Aa}$ ± 0.57 | 7.33 $^{Ba}$ ± 0.50 | 7.33 $^{Ba}$ ± 0.57 |
| | EKİ | 8.33 $^{ABa}$ ± 0.57 | 7.33 $^{Aa}$ ± 0.57 | 7.66 $^{ABa}$ ± 0.57 | 7.66 $^{ABa}$ ± 0.57 |
| | EBİ | 8.00 $^{BCa}$ ± 0.27 | 7.33 $^{Aa}$ ± 0.57 | 7.66 $^{ABa}$ ± 0.57 | 7.66 $^{ABa}$ ± 0.57 |
| | EKE | 7.66 $^{BCDa}$ ± 0.57 | 7.66 $^{Aa}$ ± 0.57 | 7.66 $^{ABa}$ ± 0.57 | 7.66 $^{ABa}$ ± 0.57 |
| Odor | G K | 8.61 $^{Aa}$ ± 0.15 | 8.66 $^{Aa}$ ± 0.22 | 8.46 $^{Aa}$ ± 0.13 | 8.46 $^{Aa}$ ± 0.27 |
| | GKİ | 8.53 $^{Aa}$ ± 0.17 | 8.66 $^{Aa}$ ± 0.57 | 8.72 $^{Aa}$ ± 0.01 | 8.33 $^{ABa}$ ± 0.57 |
| | GBİ | 6.00 $^{Ba}$ ± 1.00 | 6.66 $^{Ba}$ ± 0.57 | 7.33 $^{BCa}$ ± 0.57 | 7.33 $^{BCDa}$ ± 0.57 |
| | GKE | 6.66 $^{Ba}$ ± 0.57 | 5.33 $^{Cb}$ ± 0.57 | 6.66 $^{BCa}$ ± 0.57 | 6.66 $^{CDa}$ ± 0.57 |
| | EK | 7.00 $^{Ba}$ ± 0.17 | 7.00 $^{Ba}$ ± 1.00 | 7.66 $^{Ba}$ ± 0.57 | 7.66 $^{ABCa}$ ± 0.57 |
| | EKİ | 8.33 $^{Aa}$ ± 0.37 | 7.33 $^{Ba}$ ± 0.57 | 7.66 $^{Ba}$ ± 0.57 | 7.66 $^{ABCa}$ ± 0.57 |
| | EBİ | 6.33 $^{Ba}$ ± 1.52 | 6.33 $^{BCa}$ ± 0.57 | 6.66 $^{BCa}$ ± 0.57 | 6.66 $^{CDa}$ ± 0.57 |
| | EKE | 6.66 $^{Ba}$ ± 0.57 | 6.33 $^{Ba}$ ± 0.57 | 6.33 $^{Ca}$ ± 0.57 | 6.33 $^{Da}$ ± 0.57 |
| Taste | G K | 8.33 $^{Ab}$ ± 0.23 | 8.71 $^{Aa}$ ± 0.35 | 8.00 $^{Bb}$ ± 0.15 | 8.00 $^{Ab}$ ± 0.50 |
| | GKİ | 8.60 $^{Aa}$ ± 0.27 | 8.33 $^{ABa}$ ± 0.57 | 8.72 $^{Aa}$ ± 0.17 | 8.33 $^{Aa}$ ± 0.57 |
| | GBİ | 5.33 $^{Ba}$ ± 1.15 | 5.33 $^{CDa}$ ± 0.57 | 6.00 $^{Da}$ ± 1.00 | 6.00 $^{Ca}$ ± 1.00 |
| | GKE | 5.00 $^{Bb}$ ± 1.00 | 5.66 $^{CDab}$ ± 0.57 | 6.66 $^{CDa}$ ± 0.57 | 6.66 $^{BCa}$ ± 0.57 |
| | EK | 8.00 $^{Aa}$ ± 0.50 | 6.33 $^{Cb}$ ± 0.57 | 7.33 $^{BCa}$ ± 0.57 | 7.33 $^{ABa}$ ± 0.57 |
| | EKİ | 8.00 $^{Aa}$ ± 1.00 | 7.66 $^{Ba}$ ± 0.57 | 6.66 $^{CDa}$ ± 0.57 | 6.66 $^{BCa}$ ± 0.57 |
| | EBİ | 5.33 $^{Ba}$ ± 0.57 | 4.66 $^{Da}$ ± 0.57 | 4.66 $^{Ea}$ ± 0.57 | 4.66 $^{Da}$ ± 0.57 |
| | EKE | 6.33 $^{Ba}$ ± 0.57 | 5.66 $^{CDa}$ ± 0.57 | 6.00 $^{Da}$ ± 0.17 | 6.00 $^{Ca}$ ± 0.35 |
| Saltiness | G K | 8.00 $^{ABa}$ ± 0.56 | 7.33 $^{ABa}$ ± 0.57 | 8.46 $^{Aa}$ ± 0.14 | 8.50 $^{Aa}$ ± 0.17 |
| | GKİ | 8.42 $^{Aa}$ ± 0.35 | 8.33 $^{Aab}$ ± 0.18 | 7.66 $^{Bbc}$ ± 0.57 | 7.33 $^{Bc}$ ± 0.57 |
| | GBİ | 7.33 $^{BCab}$ ± 0.57 | 8.00 $^{Aa}$ ± 0.37 | 6.66 $^{BCb}$ ± 0.57 | 6.66 $^{Bb}$ ± 0.57 |
| | GKE | 7.66 $^{BCa}$ ± 0.57 | 7.33 $^{ABa}$ ± 0.57 | 7.33 $^{BCa}$ ± 0.57 | 7.33 $^{Ba}$ ± 0.57 |
| | EK | 8.00 $^{ABa}$ ± 0.35 | 7.33 $^{ABb}$ ± 0.57 | 7.00 $^{BCb}$ ± 0.17 | 7.00 $^{Bb}$ ± 0.37 |
| | EKİ | 7.66 $^{BCa}$ ± 0.57 | 7.66 $^{Aa}$ ± 0.57 | 7.00 $^{BCa}$ ± 0.50 | 7.00 $^{Ba}$ ± 0.17 |
| | EBİ | 6.66 $^{Ca}$ ± 0.57 | 5.66 $^{Ca}$ ± 0.57 | 6.33 $^{Ca}$ ± 0.57 | 6.33 $^{Ba}$ ± 0.57 |
| | EKE | 7.66 $^{BCa}$ ± 0.57 | 6.33 $^{BCa}$ ± 1.15 | 7.00 $^{BCa}$ ± 1.00 | 7.00 $^{Ba}$ ± 1.00 |
| Overall acceptability | G K | 8.60 $^{Aa}$ ± 0.15 | 8.33 $^{Aa}$ ± 0.51 | 8.50 $^{Aa}$ ± 0.32 | 8.45 $^{Aa}$ ± 0.15 |
| | GKİ | 8.53 $^{Aa}$ ± 0.15 | 8.53 $^{Aa}$ ± 0.27 | 8.33 $^{ABa}$ ± 0.32 | 8.73 $^{Aa}$ ± 0.25 |
| | GBİ | 5.66 $^{Ca}$ ± 0.57 | 6.00 $^{Ca}$ ± 0.17 | 6.00 $^{Ca}$ ± 1.00 | 6.00 $^{Ca}$ ± 1.00 |
| | GKE | 6.66 $^{Ba}$ ± 0.57 | 6.66 $^{BCa}$ ± 0.57 | 7.33 $^{Ba}$ ± 0.57 | 7.33 $^{Ba}$ ± 0.57 |
| | EK | 7.33 $^{Ba}$ ± 0.57 | 7.33 $^{ABa}$ ± 0.57 | 8.00 $^{ABa}$ ± 0.50 | 8.00 $^{ABa}$ ± 0.17 |
| | EKİ | 8.56 $^{Aa}$ ± 0.11 | 7.33 $^{ABb}$ ± 0.57 | 7.33 $^{Bb}$ ± 0.57 | 7.33 $^{Bb}$ ± 0.57 |
| | EBİ | 6.66 $^{Ba}$ ± 0.57 | 6.66 $^{BCa}$ ± 0.57 | 5.66 $^{Ca}$ ± 1.15 | 5.66 $^{Ca}$ ± 1.15 |
| | EKE | 7.00 $^{Ba}$ ± 0.50 | 7.00 $^{BCa}$ ± 1.00 | 7.33 $^{Ba}$ ± 0.57 | 7.33 $^{Ba}$ ± 0.57 |

**Table 6.** *Cont.*

| Parameter | Sample Code | Storage Days | | | |
|---|---|---|---|---|---|
| | | Day 1 | Day 15 | Day 30 | Day 45 |
| Purchase Intention | G K | 1.00 [Ca] ± 0.15 | 1.00 [Ba] ± 0.25 | 1.00 [Ca] ± 0.15 | 1.00 [Da] ± 0.25 |
| | GKİ | 1.00 [Ca] ± 0.32 | 1.00 [Ba] ± 0.55 | 1.00 [Ca] ± 0.25 | 1.00 [Da] ± 0.15 |
| | GBİ | 2.33 [Ba] ± 0.57 | 2.66 [Aa] ± 0.57 | 2.66 [Aa] ± 0.57 | 2.66 [ABa] ± 0.57 |
| | GKE | 2.00 [Ba] ± 0.15 | 2.33 [Aa] ± 0.57 | 2.66 [Aa] ± 0.57 | 2.66 [ABa] ± 0.57 |
| | EK | 1.33 [Ca] ± 0.43 | 1.33 [Ba] ± 0.57 | 1.66 [Ba] ± 0.57 | 1.66 [CDa] ± 0.57 |
| | EKİ | 1.00 [Cb] ± 0.15 | 1.33 [Bb] ± 0.57 | 2.00 [Ba] ± 0.15 | 2.00 [BCa] ± 0.17 |
| | EBİ | 3.00 [Aa] ± 0.50 | 3.00 [Aa] ± 0.50 | 3.00 [Aa] ± 0.25 | 3.00 [Aa] ± 0.15 |
| | EKE | 3.00 [Aa] ± 0.50 | 3.00 [Aa] ± 0.50 | 3.00 [Aa] ± 0.25 | 2.66 [ABa] ± 0.57 |

Mean ± standard deviation. **Note**: Values are expressed as **mean ± standard deviation**. **A–E**: Capital letters in the same column indicate statistically significant differences between sample types on the same storage day ($p < 0.05$). **a–c**: Lowercase letters in the same row indicate statistically significant differences within the same sample over different storage times ($p < 0.05$). **Sample Codes**: **GK**: Traditional production, control group, **GKİ**: Traditional production, Cecil cheese coated with chitosan, **GBİ**: Traditional production, Cecil cheese coated with chitosan + rosemary essential oil, **GKE**: Traditional production, Cecil cheese coated with chitosan + thyme essential oil, **EK**: Industrial production, control group, **EKİ**: Industrial production, Cecil cheese coated with chitosan, **EBİ**: Industrial production, Cecil cheese coated with chitosan + rosemary essential oil, **EKE**: Industrial production, Cecil cheese coated with chitosan + thyme essential oil.

In general, the sensory analysis results are in line with the literature. Yüceer (2017) reported higher color and texture scores in Kashar cheese coated with chitosan compared to control samples [44]. Similarly, Karakuş (2021) highlighted that cheeses coated with chitosan and essential oil-based films received higher overall acceptability scores and showed significant differences in taste characteristics [45]. Arfat et al. (2015) noted that the strong aromatic effects of essential oils added to chitosan coatings sometimes led to decreased taste and flavor acceptance among panelists [54]. This aligns with the relatively low taste and odor scores observed in rosemary oil-containing samples in the present study. Sensory evaluations revealed that traditionally produced Cecil cheeses—particularly those coated with chitosan and thyme oil—achieved higher scores in key sensory parameters such as color, texture, and taste-aroma. Essential oil content had a notable impact, especially on odor and flavor; while some panelists did not favor the aromas, others evaluated them positively for their richness. A noticeable decrease in taste and odor scores was particularly observed in groups containing rosemary oil. Nevertheless, chitosan-based films were generally effective in preserving the cheese's physical integrity and sensory quality. Panelists' purchasing tendencies highlighted the importance of maintaining the traditional form and familiar flavor profile of the product. The findings suggest that the sensory advantages of traditional production can be balanced with the protective benefits of chitosan and essential oil-based edible films, contributing positively to consumer acceptance. Sensory evaluation plays a significant role in consumer acceptance [70]. In this study, traditionally produced samples, particularly those coated with chitosan and thyme oil, received higher sensory scores. These results are consistent with the findings of Yüceer (2017) and Yangılar (2015), who reported that chitosan coatings are effective in preserving physical integrity and enhancing texture and color [44,62]. Similarly, Karakuş (2021) noted that edible films containing essential oils have a positive influence on consumer preferences [45]. However, some studies (Bleoancă et al., 2020; Tomičić et al., 2018) have pointed out that high concentrations of essential oils may negatively affect aroma and odor profiles [55,71]. This was clearly observed in the rosemary-coated EBİ sample in our study. It appears that essential oils can yield positive outcomes at low concentrations [21]; this was reflected in the GKE samples, which received acceptable aroma scores. Similar findings have been reported in the literature for coated cheeses by Vargas-Ramella et al. (2025) and Molina-Hernández et al. (2020) [56,72]. Iqbal et al. (2021) and Ressutte et al. (2022) demonstrated

that chitosan-based coatings preserve taste and texture, with sensory scores remaining within acceptable ranges [51,52]. Iqbal et al. (2021) also observed the preservation of elasticity and hardness, which aligns with the results of our study [51]. Consumer purchase intent was higher for traditionally produced and familiar products. This finding supports the studies of Kıngır and Kardeş (2019) and Burt (2004), which emphasize the impact of consumer habits on the acceptance of innovative food products [40,73]. In order to facilitate a clearer evaluation and comparison of the sensory analysis parameters of the samples, the results have been presented in graphical form in Figure 6.

3.1.5. Evaluation of Texture Analysis Results of Cecil Cheeses

When examining hardness values, the highest value on day 1 was observed in the EK sample (3695.65 g), while the lowest was found in the GKE sample (251.03 g). This indicates that the control group from industrial production exhibited a firmer structure compared to traditionally produced samples. On day 45, the highest hardness value was recorded in the EKİ group (1619.71 g), while the lowest was found in the GK sample (543.27 g). These findings suggest that the chitosan coating contributed to preserving the structural integrity of the cheese and helped reduce the loss of firmness during storage. Regarding adhesiveness, the lowest value on day 1 was found in the EK group (−8.72), indicating a less adhesive texture compared to other samples. By day 45, all samples showed a shift toward more positive adhesiveness values, with relatively low values maintained in the EKİ, EBİ, and EKE groups. This trend suggests that the coating materials may have had a regulatory effect on adhesiveness by balancing possible moisture migration. In terms of elasticity, the GKE (4.57 mm) and EKİ (4.39 mm) groups stood out on day 1. However, all samples exhibited a marked decrease in elasticity values by day 45. Notably, the increase observed in the GKİ group (1.01 mm) implies that the chitosan coating may have helped preserve the elastic properties. Although consistency values did not show significant changes between days 1 and 45, a decreasing trend was observed in the EKİ group. This suggests that while chitosan coating initially provided structural stability in industrially produced cheese, some degree of softening may have occurred over time. The gumminess and chewiness parameters reflect how long the cheese withstands mastication and the level of resistance during chewing. On day 1, the highest gumminess was found in the EK group (2790.34 g), while the highest chewiness was recorded in the EKİ group (3851.00 g). On day 45, the EKE group (1813.42 g) stood out in terms of chewiness. Regarding springiness ratio, the highest value on day 1 was observed in the EKİ group (0.56). However, by day 45, a decline of the measured parameters was noted in all groups, with particularly low values in some groups such as GBİ and GKE. This may suggest that the coating materials could not fully prevent the loss of elasticity over time (Table 7).

The textural findings obtained in this study are consistent with previous reports on the effects of similar coating applications on the physical properties of cheese. For example, Karakuş (2021) highlighted the protective effect of chitosan coating on hardness and its delaying impact on elasticity loss in Kashar cheese [45]. Similarly, Kavas and Kesenkas (2018), in their study on Cecil cheese, reported that chitosan-based coatings positively influenced chewiness and provided structural stability throughout shelf life [65]. Yüceer (2017) emphasized the textural differences between traditional and industrial cheeses, noting that industrially produced cheeses tend to have firmer and more consistent structures [44]. The present study supports this observation, as the EK and EKİ groups generally exhibited higher hardness and chewiness values.

**Table 7.** Texture analysis results of Cecil cheese samples.

| Parameter | Sample Code | Storage Days | |
| --- | --- | --- | --- |
| | | Day 1 | Day 45 |
| Hardness | G K | $1256.14^{Da} \pm 64.88$ | $543.27^{Fb} \pm 16.64$ |
| | GKİ | $680.60^{Eb} \pm 58.84$ | $1145.96^{Da} \pm 62.81$ |
| | GBİ | $734.14^{Ea} \pm 63.53$ | $755.53^{Ea} \pm 32.21$ |
| | GKE | $251.03^{Fb} \pm 15.31$ | $1103.32^{Da} \pm 76.51$ |
| | EK | $3411.68^{Aa} \pm 200.82$ | $1469.79^{BCb} \pm 12.02$ |
| | EKİ | $2396.27^{Ba} \pm 77.68$ | $2178.78^{Ab} \pm 68.83$ |
| | EBİ | $1681.39^{Ca} \pm 40.30$ | $1565.94^{Bb} \pm 38.27$ |
| | EKE | $1185.54^{Db} \pm 21.33$ | $1405.90^{Ca} \pm 13.13$ |
| Adhesiveness | G K | $-0.20^{Aa} \pm 0.05$ | $-0.66^{Cb} \pm 0.06$ |
| | GKİ | $-0.52^{ABa} \pm 0.10$ | $-0.56^{BCa} \pm 0.02$ |
| | GBİ | $-2.89^{Db} \pm 0.12$ | $-1.31^{Da} \pm 0.05$ |
| | GKE | $-2.33^{Cb} \pm 0.09$ | $-1.47^{Da} \pm 0.20$ |
| | EK | $-8.72^{Eb} \pm 0.55$ | $-1.56^{Da} \pm 0.21$ |
| | EKİ | $-0.15^{Aa} \pm 0.01$ | $-0.21^{ABa} \pm 0.04$ |
| | EBİ | $-0.29^{Ab} \pm 0.07$ | $-0.10^{Aa} \pm 0.04$ |
| | EKE | $-0.78^{Ba} \pm 0.07$ | $-1.27^{Db} \pm 0.10$ |
| Elasticity | G K | $0.99^{Ca} \pm 0.05$ | $0.97^{ABa} \pm 0.03$ |
| | GKİ | $0.87^{Cb} \pm 0.01$ | $1.01^{Aa} \pm 0.02$ |
| | GBİ | $0.88^{Cb} \pm 0.03$ | $0.96^{ABCa} \pm 0.03$ |
| | GKE | $4.57^{Aa} \pm 0.74$ | $1.05^{Ab} \pm 0.05$ |
| | EK | $0.83^{Cb} \pm 0.04$ | $0.95^{ABCa} \pm 0.01$ |
| | EKİ | $4.39^{Aa} \pm 0.39$ | $0.88^{BCb} \pm 0.01$ |
| | EBİ | $0.99^{Ca} \pm 0.02$ | $0.98^{ABa} \pm 0.03$ |
| | EKE | $2.60^{Ba} \pm 0.22$ | $0.85^{Cb} \pm 0.02$ |
| Consistency | G K | $0.73^{Da} \pm 0.02$ | $0.76^{BCa} \pm 0.02$ |
| | GKİ | $0.74^{CDb} \pm 0.02$ | $0.82^{ABa} \pm 0.01$ |
| | GBİ | $0.76^{CDb} \pm 0.01$ | $0.85^{Aa} \pm 0.01$ |
| | GKE | $0.78^{BCa} \pm 0.02$ | $0.78^{BCa} \pm 0.01$ |
| | EK | $0.79^{Ba} \pm 0.01$ | $0.78^{BCa} \pm 0.01$ |
| | EKİ | $0.84^{Aa} \pm 0.01$ | $0.79^{BCb} \pm 0.03$ |
| | EBİ | $0.74^{Db} \pm 0.01$ | $0.79^{BCa} \pm 0.03$ |
| | EKE | $0.73^{Da} \pm 0.02$ | $0.72^{Ca} \pm 0.03$ |
| Chewiness | G K | $1052.82^{Da} \pm 45.84$ | $428.97^{Eb} \pm 12.87$ |
| | GKİ | $500.26^{Eb} \pm 27.98$ | $595.14^{Da} \pm 40.93$ |
| | GBİ | $1401.94^{Ca} \pm 147.07$ | $631.05^{Db} \pm 23.47$ |
| | GKE | $345.04^{Eb} \pm 16.93$ | $791.37^{Ca} \pm 10.39$ |
| | EK | $2790.34^{Aa} \pm 207.14$ | $1177.59^{Bb} \pm 9.86$ |
| | EKİ | $1927.28^{Ba} \pm 34.16$ | $1461.04^{Ab} \pm 29.35$ |
| | EBİ | $1279.77^{CDa} \pm 80.54$ | $758.65^{Cb} \pm 37.77$ |
| | EKE | $1142.79^{CDb} \pm 21.04$ | $1420.52^{Aa} \pm 53.00$ |
| Gumminess | G K | $1330.24^{Ca} \pm 124.10$ | $416.96^{Fb} \pm 5.59$ |
| | GKİ | $466.75^{Db} \pm 10.29$ | $1246.50^{CDa} \pm 125.66$ |
| | GBİ | $1412.71^{Ca} \pm 75.04$ | $798.02^{Eb} \pm 12.63$ |
| | GKE | $1854.55^{Ba} \pm 25.45$ | $808.82^{Eb} \pm 30.01$ |
| | EK | $3695.65^{Aa} \pm 229.11$ | $1140.50^{Db} \pm 29.43$ |
| | EKİ | $3851.00^{Aa} \pm 68.57$ | $1619.71^{Bb} \pm 29.57$ |
| | EBİ | $1367.44^{Ca} \pm 38.37$ | $1323.69^{Ca} \pm 32.41$ |
| | EKE | $1543.39^{Cb} \pm 17.38$ | $1813.42^{Aa} \pm 25.22$ |

**Table 7.** *Cont.*

| Parameter | Sample Code | Storage Days | |
|---|---|---|---|
| | | Day 1 | Day 45 |
| Resilience ratio | G K | 0.52 $^{BCa}$ ± 0.02 | 0.48 $^{Db}$ ± 0.01 |
| | GKİ | 0.48 $^{CDb}$ ± 0.03 | 0.57 $^{Aa}$ ± 0.01 |
| | GBİ | 0.46 $^{DEa}$ ± 0.01 | 0.46 $^{DEa}$ ± 0.03 |
| | GKE | 0.44 $^{Eb}$ ± 0.02 | 0.46 $^{Ea}$ ± 0.03 |
| | EK | 0.51 $^{BCa}$ ± 0.01 | 0.48 $^{Db}$ ± 0.03 |
| | EKİ | 0.56 $^{Aa}$ ± 0.01 | 0.48 $^{DEb}$ ± 0.02 |
| | EBİ | 0.52 $^{Ba}$ ± 0.02 | 0.53 $^{Ba}$ ± 0.03 |
| | EKE | 0.46 $^{DEb}$ ± 0.03 | 0.51 $^{Ca}$ ± 0.07 |

Mean ± standard Deviation. **Note**: Values are expressed as **mean ± standard deviation**. **A–F**: Capital letters in the same column indicate statistically significant differences between sample types on the same storage day (*p* < 0.05). **a–b**: Lowercase letters in the same row indicate statistically significant differences within the same sample over different storage times (*p* < 0.05). **Sample Codes**: **GK**: Traditional production, control group, **GKİ**: Traditional production, Cecil cheese coated with chitosan, **GBİ**: Traditional production, Cecil cheese coated with chitosan + rosemary essential oil, **GKE**: Traditional production, Cecil cheese coated with chitosan + thyme essential oil, **EK**: Industrial production, control group, **EKİ**: Industrial production, Cecil cheese coated with chitosan, **EBİ**: Industrial production, Cecil cheese coated with chitosan + rosemary essential oil, **EKE**: Industrial production, Cecil cheese coated with chitosan + thyme essential oil.

The textural analysis results obtained in the present study are consistent with the previous literature reporting the effects of edible coatings on the textural properties of cheese. Fox et al. (2017) indicated that cheese texture is influenced by moisture loss and proteolytic changes [1]. Atarés and Chiralt (2016) and Mehdizadeh et al. (2021) demonstrated that chitosan and essential oil-based coatings act as moisture barriers, thereby slowing down the increase in hardness [11,20]. This finding aligns with the reduced softening trend observed in coated cheese samples. Regarding proteolysis control, El-Sayed and Youssef (2024) and Casalini et al. (2024) found that such coatings suppress protease activity, thus preserving structural integrity and stabilizing adhesiveness and elasticity [53,74]. This is also in agreement with our findings. Bourne (2002) stated that texture profile analysis (TPA) parameters reflect the perceived structure of cheese [75]. In our study, chitosan coatings effectively maintained these parameters. Gao et al. (2023) and Pires et al. (2024) reported that coatings containing chitosan and oregano oil significantly preserved the texture of cheese and observed reductions in hardness, gumminess, and chewiness—findings that are consistent with the favorable textural values found in the EKE and EKİ groups [57,76]. Ghasemian et al. (2024) reported that rosemary oil had no direct effect on texture but noted that its antioxidant properties may indirectly contribute to textural stability [77]. This may explain its observed effects in. Edible films containing chitosan and essential oils positively affected the textural properties during storage, particularly in industrially produced Cecil cheeses. However, the effect was more limited in traditionally produced cheeses.

The visual appearances of both traditionally and industrially produced Cecil cheese samples during the storage period are presented in Figure 7. Examination of the images revealed that the cheese samples maintained their structural integrity and did not exhibit noticeable deterioration throughout storage. From a textural perspective, the preservation of the fibrous structure and characteristic appearance of the samples indicates that the applied coating materials effectively supported the structural stability of the cheese. These findings are consistent with previous reports stating that edible coatings help preserve the textural attributes of cheese, thereby maintaining product integrity during storage [21,66]. Furthermore, the retention of visual integrity in the samples is considered an important parameter in terms of consumer acceptance.

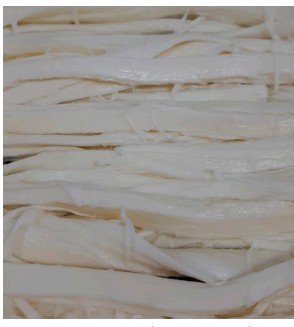

(**a**) Day 1 of storage for traditional production

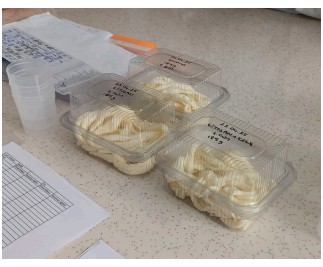

(**b**) Day 15 of storage for traditional production

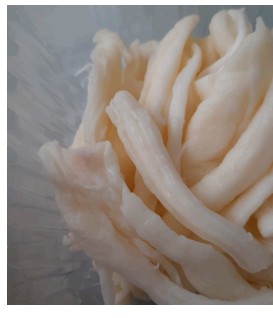

(**c**) Day 30 of storage for traditional production

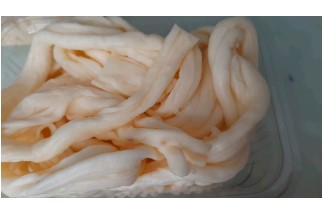

(**d**) Day 45 of storage for traditional production

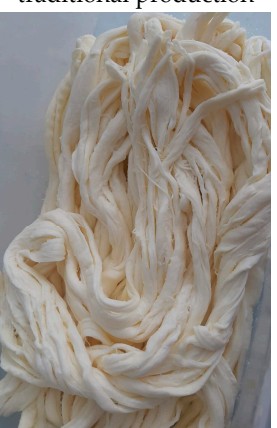

(**e**) Day 1 of storage for industrial production

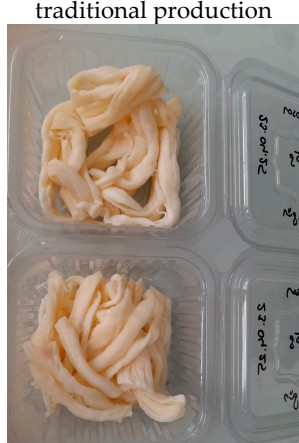

(**f**) Day 15 of storage for industrial production

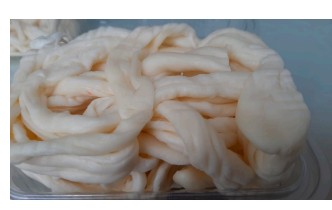

(**g**) Day 30 of storage for industrial production

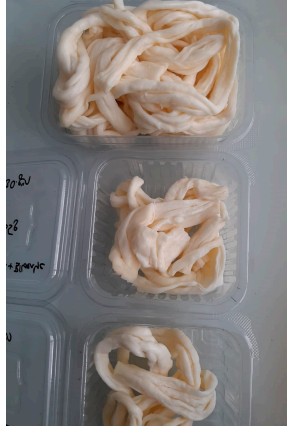

(**h**) Day 45 of storage for industrial production

**Figure 7.** Traditional and industrially produced Cecil cheese during storage days.

The texture profile parameters obtained from instrumental analyses were closely correlated with the sensory evaluation outcomes. Higher hardness, gumminess, and chewiness values of industrially produced and chitosan-coated samples were reflected in the sensory panelists' higher texture scores, particularly in the EKİ and EKE groups. Similarly, the improved elasticity and adhesiveness values in coated cheeses corresponded with panelists' perception of a more pleasant and consistent mouthfeel. In traditionally produced cheeses, despite lower instrumental hardness values, the panelists rated texture positively, likely due to the familiar and characteristic structure of the cheese. The decrease in springiness and elasticity observed over storage was consistent with slight reductions in texture scores in sensory evaluations, particularly for samples containing essential oils such as EBİ, where bitterness or strong aroma may have influenced the mouthfeel perception. These findings suggest that the protective effect of chitosan and essential oil-based edible films on moisture retention not only preserves structural integrity but also enhances the sensory perception of texture, reinforcing the importance of integrating instrumental and sensory data to fully evaluate the quality of Cecil cheese.

## 4. Conclusions

Efforts to extend the shelf life of dairy products, preserve their microbial stability, and maintain sensory quality have gained significant importance in both the academic literature and industrial applications. In this context, edible films and coatings being food-safe, environmentally friendly, biodegradable, and possessing functional properties have emerged as promising preservation systems, even for traditional food products. One of the novel aspects of this study is the comparative evaluation of quality differences between traditionally and industrially produced cheeses, along with the separate analysis

of the effects of edible film applications on each production type. Traditionally produced samples exhibited higher dry matter content from the outset, indicating differences in moisture loss due to production methods. The positive effect of edible film coatings in maintaining dry matter content was particularly evident in industrially produced samples. The well-documented antimicrobial effect of chitosan was directly observed in this study; mechanisms such as binding to microbial cell membranes, disrupting membrane permeability, and interfering with intracellular functions resulted in significant reductions in the tested microorganisms. Chitosan was thus evaluated not only as a film-forming agent but also as an active protective component. Another unique contribution of this study is the comprehensive application of edible films on a traditional cheese like Cecil cheese, which has not been previously investigated in such detail. Industrially produced cheeses, due to their more homogeneous structure, exhibited better interaction with the coatings, whereas traditionally produced cheeses retained superior sensory characteristics. Texture profile analysis confirmed that coating applications contributed to the stabilization of key parameters such as chewiness and springiness. This study provides a comprehensive evaluation of how edible film applications can improve the quality of Cecil cheese under both traditional and industrial production conditions. The results demonstrate that biopolymer- and essential oil-based films are effective in extending shelf life, preserving microbiological stability, and enhancing consumer acceptance. These findings support the applicability of natural and sustainable preservation methods in the food industry and provide strong evidence that traditional products can be adapted to modern preservation technologies. Methodological limitations include the 45-day storage period and uncertainties regarding the migration of bioactive compounds into the cheese matrix, which may affect the generalizability of the results. Future research directions should focus on applying edible films to other cheese types, investigating the migration behavior of bioactive compounds, and conducting sensory evaluations with consumer panels. These approaches are expected to contribute valuable insights to both academic research and industrial applications.

**Supplementary Materials:** The following supporting information can be downloaded at: https://www.mdpi.com/article/10.3390/fermentation11090542/s1.

**Author Contributions:** F.S.: data curation (equal), formal analysis (equal), investigation (equal), methodology (equal), project administration (equal), software (lead), validation (lead), visualization (lead), writing—original draft (lead), funding acquisition (lead). E.E.: data curation (equal), formal analysis (equal), investigation (equal), methodology (equal), visualization (supporting), writing—original draft (supporting). All authors have read and agreed to the published version of the manuscript.

**Funding:** This research received no external funding.

**Institutional Review Board Statement:** The authors have nothing to report. This study does not involve any human or animal testing.

**Data Availability Statement:** Data are available upon request.

**Acknowledgments:** This study is a part of Esra Efdal's master's thesis. Fadime Seyrekoğlu is the primary author.

**Conflicts of Interest:** The authors declare no conflict of interest.

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
