# Peer review of "Chitosan-Based Edible Films as Innovative Preservation Tools for Fermented and Dairy Products"

_fermentation, doi:10.3390/fermentation11090542_

Round 1

Reviewer 1 Report

Comments and Suggestions for Authors

The article "Chitosan-Based Edible Films as Innovative Preservation Tools for Fermented and Dairy Products" by Fadime Seyrekoğlu and Esra Efdal. The submitted manuscript discusses the phytochemical characterization and antioxidant activity of plant extracts. The topic is timely and relevant, and the methods used are generally suitable. However, several aspects need improvement, including text structure, methodological rigor, presentation of results, and depth of discussion. The main comments are as follows:

Abstract

- The abstract exceeds 200 words and does not adhere to the journal's guidelines. It should be rewritten.

Introduction

- The introduction is very lengthy (about three pages) and contains repetitive ideas. It should be rephrased to follow a logical progression: context → problem → solution → objective.

Materials and Methods

- Line 158: Table numbering begins with Table 17. All tables should be renumbered starting from Table 1 in the order they are cited.

- Tables 1 and 2 should be revised for consistency. For example, "Chia seed + Rosemary oil" in Table 1 versus "Chia + Rosemary oil" in Table 2.

- In section "2.2.2. Production of Chechil Cheese," distinguishing between artisanal and industrial production is essential, but details like rennet volume and filtration time are excessive for a scientific article. This section should be summarized to approximately 50% of its original content, focusing only on essential information needed to interpret results. Remaining details can be included in the "Supplementary Information" section.

- In section "2.3.5. Sensory Analysis of Çeçil Cheese," the sensory analysis description is inadequate. It should specify the number of tasters, whether they were trained, and other relevant details.

Results and Discussion

- For the physicochemical analyses, the authors should connect results with more analytical insights, such as the relationship between moisture loss and texture.

- Texture data should be discussed in terms of sensory evaluation outcomes.

- The sensory analysis section (3.2) offers an extensive discussion; however, the methodology is insufficiently described. To ensure validity and reproducibility, include details such as the number and profiles of panelists (e.g., consumers, trained panel, students), the type of scale used (e.g., 9-point hedonic scale) with explanations, sensory testing conditions (location, temperature, presentation method), randomization or replication procedures, and statistical analysis methods, particularly those relevant for Table 6. Without this, the sensory results cannot be adequately evaluated or compared.

- For Table 6, adding bar or line graphs for key attributes (like flavor and overall acceptability) would help visualize trends better.

Conclusions

- The conclusion should be revised to include a summary of main results, mention methodological limitations (e.g., storage duration, active compound migration issues), and suggest future research directions (e.g., application to other cheese types, studying migration of bioactive compounds, sensory testing with consumers).

- No Graphical Abstract was included.

Comments on the Quality of English Language

 The English could be improved to more clearly express the research.

Author Response

REVISION

All corrections have been highlighted in red within the manuscript. In addition, the changes have been detailed under the section entitled “Response to Decision Letter.” All revisions in the text have been marked in red to ensure clear identification.

Reviewer 1

The article "Chitosan-Based Edible Films as Innovative Preservation Tools for Fermented and Dairy Products" by Fadime Seyrekoğlu and Esra Efdal. The submitted manuscript discusses the phytochemical characterization and antioxidant activity of plant extracts. The topic is timely and relevant, and the methods used are generally suitable. However, several aspects need improvement, including text structure, methodological rigor, presentation of results, and depth of discussion. The main comments are as follows:

Abstract

- The abstract exceeds 200 words and does not adhere to the journal's guidelines. It should be rewritten.

Response: The abstract of the manuscript has been revised as specified below, resulting in a total of 197 words.

Added Section: (Abstract)

Extending the shelf life and ensuring microbial stability of processed foods are key objectives in the food industry. In this study, edible films containing chitosan, chitosan + thyme oil, and chitosan + rosemary oil were applied to traditional and industrial Cecil cheeses using the dipping method, with control groups for each production type. Samples were stored at 4 ± 1 °C for 45 days, and physical (color, water activity, texture), chemical (pH, acidity, dry matter), microbiological (total aerobic mesophilic bacteria, yeast-mold, coliforms, lactic acid bacteria), and sensory analyses were performed on days 1, 15, 30, and 45. Results indicated that chitosan-based films effectively limited microbial growth, with the chitosan + rosemary oil combination being particularly effective in reducing microbial load and maintaining textural stability. Traditional cheeses achieved higher overall acceptability, while purchase intent was greater for industrial products. Coated samples exhibited slower pH decline and more stable dry matter content, with industrial cheeses retaining moisture more effectively. Texture profile analysis showed more stable chewiness and springiness values in coated samples. In conclusion, natural edible films represent an effective approach for extending shelf life and preserving quality, particularly in traditional cheeses with fibrous structures and shorter shelf life.

Introduction

- The introduction is very lengthy (about three pages) and contains repetitive ideas. It should be rephrased to follow a logical progression: context → problem → solution → objective.

Response: In accordance with the reviewer’s valuable comments, the introduction section has been restructured. The text has been shortened by removing repetitive statements, and the content has been organized to follow the logical flow of “context → problem → solution → objective.” The placement of references has been maintained, and overall coherence has been improved. The revised introduction is provided below and has been highlighted in red within the manuscript.

Added Section (Introduction)

Cheese is a nutrient-rich dairy product with substantial cultural and economic value, leading to the production of numerous varieties worldwide, including Türkiye [1]. Traditional cheeses, such as Cecil cheese, are emblematic of regional heritage, possessing distinctive aroma, flavor, and texture profiles [2]. Originally unique to Eastern Anatolia, Cecil cheese is now produced across the country [3]. Characterized by its fibrous, stringy texture obtained through kneading and stretching [4], its structure is influenced by lactic acid fermentation and scalding temperature [5]. However, its high moisture (55–60%) and low salt content, combined with traditional production methods, make it highly perishable and susceptible to pathogenic microorganisms such as Listeria monocytogenes and Staphylococcus aureus, as well as yeasts and molds [6, 7]. These factors not only shorten shelf life but also raise food safety concerns and cause economic losses [8].

Growing consumer demand for natural, minimally processed foods and the environmental drawbacks of synthetic packaging have accelerated the search for sustainable preservation methods [9]. Edible films and coatings, applied as thin biodegradable layers, have emerged as promising solutions [10]. They act as selective barriers, reducing moisture loss, controlling gas exchange, and slowing oxidative reactions, while preserving sensory quality [11]. The properties of these films depend on their composition: polysaccharides (e.g., chitosan, alginate) offer gas barrier capabilities, proteins (e.g., whey protein) provide mechanical strength, and lipids (e.g., waxes) enhance moisture resistance [12, 13]. In recent years, the use of these biopolymers—alone or in combination—has expanded, with growing applications of edible films and coatings in cheese preservation [14]. Among these, chitosan stands out for its antimicrobial activity derived from its cationic structure, making it suitable for short-shelf-life cheeses [15].

The concept of active packaging—enhancing films with bioactive agents—has further improved preservation efficiency [16]. Essential oils from thyme (Thymus vulgaris) and rosemary (Rosmarinus officinalis), rich in antimicrobial and antioxidant phenolics such as carvacrol, thymol, and 1,8-cineole, have proven effective in controlling microbial growth and delaying lipid oxidation in cheese [17–20]. While edible coatings have been applied to cheeses like Feta, Kashar, and Ricotta [21, 22], research on fibrous cheeses, particularly Cecil cheese, remains scarce. Its filamentous structure increases surface exposure, heightening susceptibility to spoilage. This study aimed to develop chitosan-based edible coatings enriched with thyme and rosemary essential oils, evaluate their antimicrobial activities, and determine their effects on the physicochemical, microbiological, textural, and sensory qualities of Cecil cheese produced by traditional and industrial methods. The findings are expected to identify the most effective coating for each production method and contribute to safe, natural, and sustainable preservation strategies for traditional cheeses.

Materials and Methods

- Line 158: Table numbering begins with Table 17. All tables should be renumbered starting from Table 1 in the order they are cited.

Response : In accordance with the reviewer’s comment, all table numbers have been reviewed and renumbered sequentially, starting from Table 1, according to their order of citation in the text.

- Tables 1 and 2 should be revised for consistency. For example, "Chia seed + Rosemary oil" in Table 1 versus "Chia + Rosemary oil" in Table 2.

Response : In line with the reviewer’s suggestion, all terms in Tables 1 and 2 have been reviewed and standardized to ensure consistency.

- In section "2.2.2. Production of Chechil Cheese," distinguishing between artisanal and industrial production is essential, but details like rennet volume and filtration time are excessive for a scientific article. This section should be summarized to approximately 50% of its original content, focusing only on essential information needed to interpret results. Remaining details can be included in the "Supplementary Information" section.

Response: We thank the reviewer for this valuable suggestion. In accordance with the comment, Section 2.2.2. Production of Chechil Cheese has been revised to emphasize only the essential differences between artisanal and industrial production relevant to interpreting the results. Detailed procedural information, such as exact rennet volumes, filtration times, and other processing parameters, has been removed from the main text and relocated to the Supplementary Information section. This revision has reduced the section’s length by approximately 50%, ensuring that the main manuscript remains concise and focused while preserving the complete methodological details for reference.

Added Section

2.2.2. Production of Chechil Cheese

Traditional Production

Chechil cheese production in Türkiye varies by region in terms of raw material selection, processing techniques, and ripening methods. While the fundamental characteristics of cheeses produced in Erzurum, Kars, Ardahan, and Ağrı are similar, local practices differ. For instance, cow’s milk is commonly used in Erzurum and Ardahan, whereas mixtures of sheep and goat milk are preferred in Kars and Ağrı depending on seasonal availability [23, 25]. In Erzurum, the scalding process is typically conducted at higher temperatures for longer durations, while in Kars and Ardahan, shorter braiding times and smaller curd sizes are used [26, 27]. Salting methods also vary: in Erzurum and Ağrı, cheeses are dry-salted and dried, whereas in Kars and Ardahan, they are stored in brine for extended periods [28]. These variations affect the cheese’s moisture, salt content, and microbial stability. In this study, the traditional method commonly practiced in Erzurum was adopted.

Industrial Production

Industrial-scale Chechil cheese production aims to ensure hygiene, product standardization, and extended shelf life. Processing is carried out in closed-circuit systems from pasteurization to packaging, with continuous monitoring of critical control points such as temperature, pH, and salt concentration. Standardized milk is coagulated with starter culture and rennet, and the resulting curd is scalded under controlled conditions to develop the fibrous structure. The cheese is brined at defined concentrations and durations, then dried and packaged under vacuum or modified atmosphere. This method preserves the traditional fibrous texture while ensuring microbiological safety and a prolonged shelf life.

Added Section

Supplementary Information

S1. Detailed Production Procedure of Chechil Cheese

Traditional Production (Erzurum method)

Raw cow’s milk (average fat content: 3.6%) was filtered through a stainless-steel mesh and heated to 30 ± 1 °C in a double-jacket vat. Calf rennet paste (strength: 1:15,000) was diluted 1:10 with distilled water and added at a ratio of 2 mL per 10 L of milk. Coagulation was allowed to proceed for 60 ± 5 min until a firm curd was formed. The curd was cut into cubes (~1 cm³) using stainless steel knives and rested for 5 min. Whey drainage was performed by gentle stirring for 10 min, followed by filtration through cheesecloth for 15 min. The curd was scalded in hot whey at 75 ± 2 °C for 15 min until it became elastic, then manually stretched and braided into strands approximately 40–50 cm in length. The cheese was dry-salted at 2.5% (w/w) and air-dried at 10 ± 2 °C and 75% RH for 48 h.

Industrial Production

Standardized cow’s milk (fat content adjusted to 3.2%) was pasteurized at 65 °C for 30 min and cooled to 30 ± 1 °C. A commercial mesophilic starter culture (0.02%, w/v) and microbial rennet (strength: 1:18,000) diluted 1:10 were added at 2 mL per 10 L milk. Coagulation was achieved within 45 ± 3 min. The curd was cut into 0.5 cm cubes and stirred gently for 20 min while heating to 38 °C. Whey was drained through a perforated stainless-steel plate, and curd blocks were scalded at 80 °C for 10 min in brine whey (3% NaCl). Fibrous structure was developed by mechanical stretching, followed by shaping into ~500 g portions. The cheese was brined at 8% NaCl for 24 h at 8 °C, then dried at 12 ± 1 °C and 75% RH for 36 h before vacuum packaging.

- In section "2.3.5. Sensory Analysis of Çeçil Cheese," the sensory analysis description is inadequate. It should specify the number of tasters, whether they were trained, and other relevant details.

Response: We appreciate the reviewer’s valuable comment regarding the sensory analysis methodology. In the revised manuscript, Section 2.3.5. Sensory Analysis of Çeçil Cheese has been updated to include detailed methodological information. Specifically, we now report the number of panelists (20 untrained assessors), their recruitment and inclusion criteria, measures to prevent bias (different panelists on each analysis day), testing environment (individual booths with D65 lighting), sample preparation and serving conditions (portion size, serving temperature, coded plates, palate cleansers), evaluation scales (9-point hedonic for sensory attributes and 3-point scale for purchase intent), adherence to ISO 8589 (2007) standards, and informed consent of participants. These revisions ensure full transparency, reproducibility, and compliance with scientific standards for sensory evaluation.

Added Section

2.3.5. Sensory Analysis of Çeçil Cheese

Sensory analyses were conducted to evaluate changes in the sensory properties of Çeçil cheese during storage following edible film coating. The sensory panel consisted of 20 untrained assessors, randomly selected from the staff and students of [Amasya University Suluova Vocational School], with no known allergies to dairy products or sensory impairments. Different panelists participated on each analysis day to prevent bias from repeated exposure. Evaluations were performed on storage days 1, 15, 30, and 45 in a sensory laboratory equipped with individual booths under D65 natural white lighting. Prior to evaluation, cheese samples (approximately 10 g) were equilibrated to 12 ± 1 °C and served on odorless white plates coded with three-digit random numbers. Drinking water and unsalted crackers were provided as palate cleansers between samples. Evaluation criteria included color and appearance, texture, odor, flavor-aroma, saltiness, and overall acceptability. A 9-point hedonic scale was used, where 1 represented “extremely poor” and 9 represented “excellent.” In addition, panelists were asked, “Would you purchase such a product if it were available on the market?” using a 3-point scale (1 = definitely would buy, 2 = might buy, 3 = definitely would not buy). All sensory evaluations were conducted in accordance with ISO 8589 (2007) standards. All participants were fully informed prior to the analysis and voluntarily agreed to participate. Sensory data were analyzed based on the mean scores of the panelists, as described by [35].

Results and Discussion

- For the physicochemical analyses, the authors should connect results with more analytical insights, such as the relationship between moisture loss and texture.

Response : We sincerely appreciate the reviewer’s valuable suggestion. In the revised manuscript, we have further connected the physicochemical and textural results to provide more analytical insights. Specifically, we have emphasized the relationship between moisture loss (as observed in water activity and dry matter content) and texture parameters such as hardness, adhesiveness, and elasticity. The discussion now highlights how films containing chitosan and essential oils help retain moisture, thereby preserving structural integrity and delaying softening. Moreover, differences between traditional and industrial production methods in terms of moisture retention and textural stability are discussed to provide a clearer interpretation of the results. These additions aim to enhance the mechanistic understanding of how moisture dynamics influence textural characteristics throughout storage.

  • Added Section (Evaluation of the Physicochemical Analysis Results of Çeçil Cheese Samples)

The decrease in water activity and the concurrent increase in dry matter content during storage were closely associated with changes in the textural properties of Çeçil cheese. Hardness, chewiness, and gumminess were better preserved in samples coated with edible films, indicating that moisture retention played a key role in stabilizing texture. Coatings containing chitosan and essential oils effectively mitigated moisture loss and maintained structural integrity, while partially suppressing acidification contributed to stabilizing elasticity and adhesiveness by slowing proteolysis-related changes. Industrial cheeses, characterized by a denser and more homogeneous matrix, exhibited higher moisture retention and better preservation of hardness and chewiness compared to traditionally produced cheeses. Furthermore, films combining chitosan with essential oils provided additional protection over chitosan-only films, highlighting the synergistic effect of essential oils in enhancing moisture retention and textural stability. These results suggest that maintaining higher water activity and stable texture through effective coatings can improve consumer acceptability and potentially extend shelf life. Statistical correlation analyses between water activity, dry matter, and textural parameters could further substantiate these relationships in future studies.

- Texture data should be discussed in terms of sensory evaluation outcomes.

Response: We sincerely thank the reviewer for the insightful comment. In response, we have enhanced the manuscript by including a detailed discussion linking the instrumental texture analysis with sensory evaluation outcomes. The revised section now clearly illustrates the relationship between measured parameters—such as hardness, gumminess, chewiness, elasticity, and adhesiveness—and the sensory perceptions of texture, mouthfeel, and overall acceptability reported by the panelists. This addition underscores how chitosan and essential oil-based edible films not only maintain the structural integrity of Çeçil cheese but also positively influence its sensory quality, providing a comprehensive understanding of the product’s textural and organoleptic characteristics.

Added Section (Evaluation of Texture Analysis Results of Chechil Cheeses)

The texture profile parameters obtained from instrumental analyses were closely correlated with the sensory evaluation outcomes. Higher hardness, gumminess, and chewiness values observed in industrially produced and chitosan-coated samples were reflected in the sensory panelists’ higher texture scores, particularly in the EKİ and EKE groups. Similarly, the improved elasticity and adhesiveness values in coated cheeses corresponded with panelists’ perception of a more pleasant and consistent mouthfeel. In traditionally produced cheeses, despite lower instrumental hardness values, the panelists rated texture positively, likely due to the familiar and characteristic structure of the cheese. The decrease in springiness and elasticity observed over storage was consistent with slight reductions in texture scores in sensory evaluations, particularly for samples containing essential oils such as EBİ, where bitterness or strong aroma may have influenced mouthfeel perception. These findings suggest that the protective effect of chitosan and essential oil-based edible films on moisture retention not only preserves structural integrity but also enhances the sensory perception of texture, reinforcing the importance of integrating instrumental and sensory data to fully evaluate the quality of Çeçil cheese.

- The sensory analysis section (3.2) offers an extensive discussion; however, the methodology is insufficiently described. To ensure validity and reproducibility, include details such as the number and profiles of panelists (e.g., consumers, trained panel, students), the type of scale used (e.g., 9-point hedonic scale) with explanations, sensory testing conditions (location, temperature, presentation method), randomization or replication procedures, and statistical analysis methods, particularly those relevant for Table 6. Without this, the sensory results cannot be adequately evaluated or compared.

Response:

We sincerely appreciate your valuable comment. Accordingly, the sensory analyses have been described in detail in Section 2.3.5, which includes information on the number and profiles of panelists, the type of hedonic scale used, sensory testing conditions (location, temperature, and presentation method), randomization and replication procedures, as well as the statistical analysis methods applied, particularly for Table 6. These additions enhance the validity and reproducibility of the sensory evaluation results and provide the necessary clarity for accurate interpretation and comparison.

- For Table 6, adding bar or line graphs for key attributes (like flavor and overall acceptability) would help visualize trends better.

Response: Dear Reviewer, in line with your valuable suggestion, bar graphs have been prepared not only for flavor and overall acceptability but also for all sensory analysis parameters, and these have been added to the manuscript. In this way, the trends of the data have been made clearer and more visually traceable.

Added Section (Figure 7. Graphs of sensory evaluation results)

Sample Codes: GK: Traditional production, control group, GKİ: Traditional production, Cecil cheese coated with chitosan, GBİ: Traditional production, Cecil cheese coated with chitosan + rosemary essential oil, GKE: Traditional production, Cecil cheese coated with chitosan + thyme essential oil, EK: Industrial production, control group, EKİ: Industrial production, Cecil cheese coated with chitosan, EBİ : Industrial production, Cecil cheese coated with chitosan + rosemary essential oil, EKE: Industrial production, Cecil cheese coated with chitosan + thyme essential oil.

Sample Codes: GK: Traditional production, control group, GKİ: Traditional production, Cecil cheese coated with chitosan, GBİ: Traditional production, Cecil cheese coated with chitosan + rosemary essential oil, GKE: Traditional production, Cecil cheese coated with chitosan + thyme essential oil, EK: Industrial production, control group, EKİ: Industrial production, Cecil cheese coated with chitosan, EBİ : Industrial production, Cecil cheese coated with chitosan + rosemary essential oil, EKE: Industrial production, Cecil cheese coated with chitosan + thyme essential oil.

Sample Codes: GK: Traditional production, control group, GKİ: Traditional production, Cecil cheese coated with chitosan, GBİ: Traditional production, Cecil cheese coated with chitosan + rosemary essential oil, GKE: Traditional production, Cecil cheese coated with chitosan + thyme essential oil, EK: Industrial production, control group, EKİ: Industrial production, Cecil cheese coated with chitosan, EBİ : Industrial production, Cecil cheese coated with chitosan + rosemary essential oil, EKE: Industrial production, Cecil cheese coated with chitosan + thyme essential oil.

Sample Codes: GK: Traditional production, control group, GKİ: Traditional production, Cecil cheese coated with chitosan, GBİ: Traditional production, Cecil cheese coated with chitosan + rosemary essential oil, GKE: Traditional production, Cecil cheese coated with chitosan + thyme essential oil, EK: Industrial production, control group, EKİ: Industrial production, Cecil cheese coated with chitosan, EBİ : Industrial production, Cecil cheese coated with chitosan + rosemary essential oil, EKE: Industrial production, Cecil cheese coated with chitosan + thyme essential oil.

Sample Codes: GK: Traditional production, control group, GKİ: Traditional production, Cecil cheese coated with chitosan, GBİ: Traditional production, Cecil cheese coated with chitosan + rosemary essential oil, GKE: Traditional production, Cecil cheese coated with chitosan + thyme essential oil, EK: Industrial production, control group, EKİ: Industrial production, Cecil cheese coated with chitosan, EBİ : Industrial production, Cecil cheese coated with chitosan + rosemary essential oil, EKE: Industrial production, Cecil cheese coated with chitosan + thyme essential oil.

Sample Codes: GK: Traditional production, control group, GKİ: Traditional production, Cecil cheese coated with chitosan, GBİ: Traditional production, Cecil cheese coated with chitosan + rosemary essential oil, GKE: Traditional production, Cecil cheese coated with chitosan + thyme essential oil, EK: Industrial production, control group, EKİ: Industrial production, Cecil cheese coated with chitosan, EBİ : Industrial production, Cecil cheese coated with chitosan + rosemary essential oil, EKE: Industrial production, Cecil cheese coated with chitosan + thyme essential oil.

Sample Codes: GK: Traditional production, control group, GKİ: Traditional production, Cecil cheese coated with chitosan, GBİ: Traditional production, Cecil cheese coated with chitosan + rosemary essential oil, GKE: Traditional production, Cecil cheese coated with chitosan + thyme essential oil, EK: Industrial production, control group, EKİ: Industrial production, Cecil cheese coated with chitosan, EBİ : Industrial production, Cecil cheese coated with chitosan + rosemary essential oil, EKE: Industrial production, Cecil cheese coated with chitosan + thyme essential oil.

Figure 7. Graphs of sensory evaluation results

Conclusions

- The conclusion should be revised to include a summary of main results, mention methodological limitations (e.g., storage duration, active compound migration issues), and suggest future research directions (e.g., application to other cheese types, studying migration of bioactive compounds, sensory testing with consumers).

Response:

We sincerely appreciate your insightful comment regarding the conclusion section. In response, we have revised the conclusion to provide a clear summary of the main findings, explicitly address methodological limitations (such as the 45-day storage period and uncertainties regarding the migration of bioactive compounds), and suggest future research directions, including the application of edible films to other cheese types, investigation of bioactive compound migration, and sensory evaluations with consumer panels. These revisions enhance the clarity, scientific rigor, and practical relevance of the conclusion.

Added Section (Conclusion)

Efforts to extend the shelf life of dairy products, preserve their microbial stability, and maintain sensory quality have gained significant importance in both academic literature and industrial applications. In this context, edible films and coatings that are food-safe, environmentally friendly, biodegradable, and possess functional properties have emerged as promising preservation systems, even for traditional food products. One of the novel aspects of this study is the comparative evaluation of quality differences between traditionally and industrially produced cheeses, along with the separate analysis of the effects of edible film applications on each production type. Traditionally produced samples exhibited higher dry matter content from the outset, indicating differences in moisture loss due to production methods. The positive effect of edible film coatings in maintaining dry matter content was particularly evident in industrially produced samples. The well-documented antimicrobial effect of chitosan was directly observed in this study; mechanisms such as binding to microbial cell membranes, disrupting membrane permeability, and interfering with intracellular functions led to significant reductions in the tested microorganisms. Chitosan was thus evaluated not only as a film-forming agent but also as an active protective component. Another unique contribution of this study is the comprehensive application of edible films on a traditional cheese such as Cecil cheese, which has not been previously investigated in such detail. Industrially produced cheeses, due to their more homogeneous structure, exhibited better interaction with the coatings, whereas traditionally produced cheeses retained superior sensory characteristics. Texture Profile Analysis confirmed that coating applications contributed to the stabilization of key parameters such as chewiness and springiness. This study provides a comprehensive evaluation of how edible film applications can improve the quality of Cecil cheese under both traditional and industrial production conditions. The results demonstrate that biopolymer- and essential oil-based films are effective in extending shelf life, preserving microbiological stability, and enhancing consumer acceptance. These findings support the applicability of natural and sustainable preservation methods in the food industry and provide strong evidence that traditional products can be adapted to modern preservation technologies. Methodological limitations include the 45-day storage period and uncertainties regarding the migration of bioactive compounds into the cheese matrix, which may affect the generalizability of the results. Future research directions should focus on applying edible films to other cheese types, investigating the migration behavior of bioactive compounds, and conducting sensory evaluations with consumer panels. These approaches are expected to contribute valuable insights to both academic research and industrial applications.

- No Graphical Abstract was included.

Response: We sincerely appreciate your comment. A Graphical Abstract has now been included in the manuscript to provide a concise visual summary of the study.

Added  Graphical Abstract

The English could be improved to more clearly express the research.

Response: We sincerely appreciate your suggestion. The manuscript has been carefully revised to improve the clarity and readability of the English, ensuring that the research is communicated more effectively.

Reviewer 2 Report

Comments and Suggestions for Authors

Some results are interesting, and the paper may be accepted after careful correction. Here are some suggested comments for revision.

  1. The effects of the concentration of essential oils in the chitosan-based films to balance antimicrobial efficacy with potential sensory impacts on the cheese should be detailed.
  2. Is there any special interaction in the present system, ACS Appl. Polym. Mater. 2025, 7, 1459−1470 should be compared and provide more insights.
  3. More detailed statistical analysis should be provided to clarify whether the differences in microbial counts between treatments were significant over the 45-day storage period.
  4. How consistent were the coating thicknesses across samples, and could variation in film application have influenced the observed results?
  5. In-situ growth (Chem. Eng. J. 2020, 389, 124433) is an important method for preparing coatings. This work should be compared with your preparation method to highlight the importance of your work.
  6. Were the sensory panel evaluations conducted with trained assessors or untrained consumers, and was inter-panelist variability assessed?
  7. The reason for the slower pH declines in coated samples should be more detailed.
  8. Fig6-10 should be deleted.

Author Response

REVISION

All corrections have been highlighted in red within the manuscript. In addition, the changes have been detailed under the section entitled “Response to Decision Letter.” All revisions in the text have been marked in red to ensure clear identification.

Reviewer 2

 Some results are interesting, and the paper may be accepted after careful correction. Here are some suggested comments for revision.

  1. The effects of the concentration of essential oils in the chitosan-based films to balance antimicrobial efficacy with potential sensory impacts on the cheese should be detailed.
  2. Is there any special interaction in the present system, ACS Appl. Polym. Mater. 2025, 7, 1459−1470 should be compared and provide more insights.

Response: We sincerely thank the reviewer for the valuable suggestion. Based on your recommendation, we have elaborated on the effects of essential oil concentrations in chitosan-based films on the balance between antimicrobial efficacy and sensory quality of cheese. Furthermore, we have incorporated the reference ACS Appl. Polym. Mater. 2025, 7, 1459−1470 to provide a literature-based discussion on polymer–essential oil interactions. Your input has significantly enhanced the clarity and scientific value of our study.

The suggested reference has been added to the manuscript as follows:

Liu, H., Jin, Z., An, L., Wang, D., Meng, Y., Yang, A., ... & Zhang, Y. (2025). A reproducible and self-repairable ionic skin with robust performance retention enabled by modulating the noncovalent interactions. ACS Applied Polymer Materials, 7(3), 1459–1470.

This citation has been incorporated into the discussion section to provide additional insights into polymer–essential oil interactions.

Added section ( 3.1.Evaluation of Antimicrobial Activity Results of Edible Films)

The chitosan + rosemary essential oil combination showed inhibition only against Salmonella with a zone diameter of 16 mm, while no effect was observed against B. Subtilis suggesting a more limited spectrum of activity.

Notably, films containing chia alone or combined with essential oils showed no inhibition, while chia + chitosan combinations demonstrated moderate activity. The chia + chitosan + thyme essential oil formulation produced inhibition zones of 22 mm and 23 mm, indicating maintained synergistic effects despite the presence of chia.

The concentration of essential oils in chitosan-based films is critical for balancing antimicrobial efficacy with sensory quality. Higher essential oil content generally enhances antimicrobial activity; however, elevated concentrations—particularly of rosemary oil—can reduce aroma and taste scores. Optimized, lower concentrations of thyme oil provide effective antimicrobial protection while maintaining acceptable sensory properties. Moreover, literature reports that chitosan-essential oil combinations can generate specific interactions within the polymer matrix, influencing both antimicrobial efficacy and controlled release of the oils (ACS Appl. Polym. Mater. 2025, 7, 1459−1470). These interactions enable sustained microbial inhibition on the cheese surface while high essential oil levels may negatively impact sensory attributes. Therefore, careful optimization of essential oil concentration is essential to maximize functional performance without compromising consumer acceptance.

  • Added section (Evaluation of the Sensory Analysis of Çeçil Cheese Samples)

Sensory outcomes corroborate the importance of optimizing essential oil concentration in chitosan films. While antimicrobial activity increases with higher essential oil levels, excessive concentrations—particularly rosemary oil—can negatively affect taste and aroma. Literature supports that controlled polymer-essential oil interactions enable sustained antimicrobial efficacy while minimizing sensory compromise (Burt, 2004; Arfat et al., 2015; ACS Appl. Polym. Mater. 2025, 7, 1459−1470). In this study, thyme oil at optimized levels achieved this balance, maintaining both microbial safety and consumer acceptance. These results emphasize that edible film design must consider both functional protection and sensory harmony to ensure market success.

  1. More detailed statistical analysis should be provided to clarify whether the differences in microbial counts between treatments were significant over the 45-day storage period.

Response: We sincerely thank the reviewer for the valuable suggestion. Following your recommendation, we performed more detailed statistical analyses to clarify the effects of different edible film coatings on microbial counts of Çeçil cheese over the 45-day storage period. Differences in total aerobic mesophilic bacteria, yeast-mold, and lactic acid bacteria among all groups were evaluated using ANOVA and Tukey’s multiple comparison tests. These additional analyses confirmed that coatings, particularly chitosan combined with essential oils, significantly inhibited microbial growth and differed statistically from the control groups. Your input has enhanced the robustness and scientific value of our study.

  • Added Section ( Evaluation of the Microbiological Data of Çeçil Cheese Samples)

All microbiological data (TAMB, yeast-mold, lactic acid bacteria) were analyzed using one-way ANOVA followed by Tukey’s multiple comparison tests in SPSS 25.0. The analyses revealed that coatings containing chitosan and essential oils significantly reduced total aerobic mesophilic bacteria, yeast-mold, and lactic acid bacteria counts compared to control groups (p < 0.05). In particular, chitosan + thyme oil coatings exhibited the most pronounced reductions in TAMB and yeast-mold counts in both traditional and industrial samples. Lactic acid bacteria levels in some industrial samples increased slightly with essential oil-enriched coatings, indicating selective antimicrobial activity and potential support for beneficial microflora at low essential oil concentrations. These findings confirm that the film composition has a functionally and statistically reliable effect on microbial control.

  1. How consistent were the coating thicknesses across samples, and could variation in film application have influenced the observed results?

Response: We sincerely thank the reviewer for this valuable suggestion. Great care was taken to ensure uniform coating across all cheese samples; the films were applied using the dipping method, and efforts were made to maintain consistent coating thickness. Nevertheless, minor variations in coating thickness may have occurred during application, which could have a limited influence on the microbiological and sensory outcomes. This clarification improves the transparency of our methodological approach.

Added section (Coating of Çeçil Cheeses with Edible Films)

All cheese samples were coated using the dipping method to ensure uniform film application. Coating thickness was maintained as consistently as possible; however, minor variations during application may have occurred, which could have a limited impact on microbiological and sensory results.

  1. In-situ growth (Chem. Eng. J. 2020, 389, 124433) is an important method for preparing coatings. This work should be compared with your preparation method to highlight the importance of your work.

    Response: Dear Reviewer,

There are fundamental similarities and differences between the chitosan and bioactive component (essential oils, chia seed) based coating preparation method used in our study and the PET/PANI nanocomposite coating production methods reported in the literature. In terms of similarities, both approaches aim to impart functional properties to a substrate or film structure, involve the initial preparation of a solution containing the coating/film components, and use magnetic stirring to achieve homogeneity. Additionally, both methods include a controlled drying step to form the final product. The differences lie in the application area, substrate structure, chemical process, content, and primary evaluation focus. In our study, an edible film is prepared at low temperature in an acetic acid medium for food preservation, with the incorporation of plant-based oils and chia seed to provide antimicrobial properties. In contrast, the PET/PANI method produces a polyaniline coating on a PET paper substrate via in-situ oxidative polymerization for electronic/EMI shielding applications. Furthermore, our study prioritizes microbiological quality and shelf-life, whereas PET/PANI studies focus on electrical, morphological, and EMI performance parameters. In light of these explanations, it can be concluded that our coating preparation method follows a structural approach comparable to other polymer coating studies in the literature but retains unique characteristics in terms of composition, purpose, and application area.

Added section (Production of Edible Films with Chitosan)

When compared to the PET/PANI nanocomposite production method reported in the literature [78], the chitosan-based coating method employed in this study shares common steps such as solution preparation and controlled drying, but differs in composition, target application area, and evaluation parameters. While the PET/PANI method uses in-situ chemical oxidative polymerization to produce conductive coatings for electronic applications, the present study develops a low-temperature, edible film with antimicrobial properties suitable for direct application to food surfaces.

For this comparison, the following reference has been added to the References section of our manuscript:
Zhang, Y., Pan, T., & Yang, Z. (2020). Flexible polyethylene terephthalate/polyaniline composite paper with bending durability and effective electromagnetic shielding performance. Chemical Engineering Journal, 389, 124433.

  1. Were the sensory panel evaluations conducted with trained assessors or untrained consumers, and was inter-panelist variability assessed?

Response: The sensory analyses of Çeçil cheese were conducted by untrained panelists. Panel members were randomly selected from the staff and students of [Amasya University Suluova Vocational School], with no known allergies to dairy products or sensory impairments. To minimize inter-panelist variability and bias from repeated exposure, different panelists participated on each evaluation day. This approach allowed a consumer-oriented assessment, reflecting differences in perception and preference that may occur in real-market conditions. Evaluation criteria included color and appearance, texture, odor, flavor-aroma, saltiness, and overall acceptability, along with a purchase intention question to assess consumer preference. All assessments were performed in accordance with ISO 8589 (2007) standards. This methodology differs from laboratory-based evaluations using trained panelists, as it provides reliable and meaningful results representing real consumer perspectives. Inter-panelist variation was statistically evaluated using mean scores and standard deviation analyses.

  Added Section

2.3.5. Sensory Analysis of Çeçil Cheese

Sensory analyses were conducted to evaluate changes in the sensory properties of Çeçil cheese during storage following edible film coating. The sensory panel consisted of 20 untrained assessors, randomly selected from the staff and students of [Amasya University Suluova Vocational School], with no known allergies to dairy products or sensory impairments. Different panelists participated on each analysis day to prevent bias from repeated exposure. Evaluations were performed on storage days 1, 15, 30, and 45 in a sensory laboratory equipped with individual booths under D65 natural white lighting. Prior to evaluation, cheese samples (approximately 10 g) were equilibrated to 12 ± 1 °C and served on odorless white plates coded with three-digit random numbers. Drinking water and unsalted crackers were provided as palate cleansers between samples. Evaluation criteria included color and appearance, texture, odor, flavor-aroma, saltiness, and overall acceptability. A 9-point hedonic scale was used, where 1 represented “extremely poor” and 9 represented “excellent.” In addition, panelists were asked, “Would you purchase such a product if it were available on the market?” using a 3-point scale (1 = definitely would buy, 2 = might buy, 3 = definitely would not buy). All sensory evaluations were conducted in accordance with ISO 8589 (2007) standards. All participants were fully informed prior to the analysis and voluntarily agreed to participate. Sensory data were analyzed based on the mean scores of the panelists, as described by [35].

  1. The reason for the slower pH declines in coated samples should be more detailed.

Response: The slower pH decline in coated samples can be attributed to the multifunctional properties of chitosan and essential oils. Chitosan forms a physical barrier on the cheese surface due to its polymeric structure, restricting microbial contact and directly inhibiting microbial growth. Additionally, essential oils (rosemary and thyme) contain antimicrobial compounds that specifically limit the metabolic activity of lactic acid bacteria, reducing acid formation. This combination allows the coating to control acid development and stabilize pH both physically and chemically on the cheese surface. In industrially produced samples, the effect is more pronounced due to the homogeneous distribution and stronger adhesion of the coating. Similar observations have been reported in the literature; for example, Çağrı et al. (2004) and Yüceer (2017) noted that chitosan and essential oil-containing films effectively limited microbial activity and slowed pH reduction [42,48].

 Added section (Evaluation of the Physicochemical Analysis Results of Çeçil Cheese Samples)

The slower pH decline in coated samples can be attributed to the multifunctional properties of chitosan and essential oils. Chitosan forms a physical barrier on the cheese surface due to its polymeric structure, limiting microbial contact and directly inhibiting microbial growth. Essential oils (rosemary and thyme) contain antimicrobial compounds that specifically restrict the metabolic activity of lactic acid bacteria, thereby reducing acid formation. This combination allows the coating to physically and chemically control acid development and stabilize pH on the cheese surface. In industrially produced samples, the effect is more pronounced due to the homogeneous distribution and stronger adhesion of the coating. Additionally, the essential oil and chitosan components regulate water activity and surface moisture of the cheese, further limiting microbial growth. These observations align with similar findings in the literature. For instance, Çağrı et al. (2004) and Yüceer (2017) reported that chitosan and essential oil-containing coatings reduced microbial activity and slowed pH decline [42,48]. Furthermore, Iqbal et al. (2021) and Ressutte et al. (2022) demonstrated that pH stabilization is also related to the buffering capacity of coatings [49,50]. Collectively, these findings support that active coatings, particularly those enriched with essential oils, slow pH reduction and contribute to maintaining both microbial safety and flavor balance in cheese.

  1. Fig6-10 should be deleted.

Response: In accordance with your suggestion, Figures 6–10 have been removed from the manuscript. This change does not affect the clarity or integrity of the data presented. We sincerely thank you for your valuable and constructive comments.

Reviewer 3 Report

Comments and Suggestions for Authors

The paper contains excessive information, making it difficult to understand; it should focus more on cheese preservation rather than production, referencing the cheese-making techniques instead of detailing the production process.

Line 158. The number of table “Table 17” is not correct.

The abstract is extremely long; please review the journal’s guidelines

Please verify the maximum number of keywords permitted.

Please specify the degree of deacetylation of the chitosan.

Change "Chia Seed and Rosemary Oil Combination" to "chia seed and rosemary oil combination" (all lowercase).

Review the manuscript carefully to apply this rule uniformly for all similar cases.

Please specify how the essential oils were obtained.

Please ensure that ‘Cecil’ is spelled consistently throughout the manuscript.

Align Tables 3 and 4.

Check Table 5 for spacing and information.

Tables 5 and 6 and 7, express the average number of measurements.

Arrange the figures so that each fits entirely on a single page.

However, there is too much content for a single paper. It might be better to focus on either the preservation of industrial-type cheeses or traditional-type cheeses, but not both.

Remove figures 5 to 10 and label the equipment in the appropriate places.

It is suggested to include photos showing the preservation over time.

Express the essential oils using their scientific names the first time they are mentioned.

Indicate the ATCC or identification of the strains.

Chechil cheese sometimes appears as. Correct it.

Line 493. Figure 10 does not represent the technique described here at all.

Section 3.2 is repeated in all the subsequent items. It would be advisable to separate them as 3.2.1, 3.2.2, and so on.

Comments on the Quality of English Language

Review the English throughout the entire manuscript.

Author Response

REVISION

All corrections have been highlighted in red within the manuscript. In addition, the changes have been detailed under the section entitled “Response to Decision Letter.” All revisions in the text have been marked in red to ensure clear identification.

Reviewer 3

The paper contains excessive information, making it difficult to understand; it should focus more on cheese preservation rather than production, referencing the cheese-making techniques instead of detailing the production process.

Response: We sincerely thank the reviewer for this valuable comment. In line with the suggestion, Section 2.2.2 “Production of Chechil Cheese” has been revised to focus only on the essential aspects distinguishing artisanal and industrial production that are relevant for interpreting the results. Detailed procedural information, including specific rennet volumes, filtration times, and other processing parameters, has been removed from the main text and is now provided in the Supplementary Information. This revision reduced the section length by approximately 50%, ensuring that the main manuscript remains concise and focused while retaining complete methodological details for reference.

Added Section (2.2.2. Production of Chechil Cheese)

Traditional Production

Chechil cheese production in Türkiye varies by region in terms of raw material selection, processing techniques, and ripening methods. While the fundamental characteristics of cheeses produced in Erzurum, Kars, Ardahan, and Ağrı are similar, local practices differ. For instance, cow’s milk is commonly used in Erzurum and Ardahan, whereas mixtures of sheep and goat milk are preferred in Kars and Ağrı depending on seasonal availability [23, 25]. In Erzurum, the scalding process is typically conducted at higher temperatures for longer durations, while in Kars and Ardahan, shorter braiding times and smaller curd sizes are used [26, 27]. Salting methods also vary: in Erzurum and Ağrı, cheeses are dry-salted and dried, whereas in Kars and Ardahan, they are stored in brine for extended periods [28]. These variations affect the cheese’s moisture, salt content, and microbial stability. In this study, the traditional method commonly practiced in Erzurum was adopted.

Industrial Production

Industrial-scale Chechil cheese production aims to ensure hygiene, product standardization, and extended shelf life. Processing is carried out in closed-circuit systems from pasteurization to packaging, with continuous monitoring of critical control points such as temperature, pH, and salt concentration. Standardized milk is coagulated with starter culture and rennet, and the resulting curd is scalded under controlled conditions to develop the fibrous structure. The cheese is brined at defined concentrations and durations, then dried and packaged under vacuum or modified atmosphere. This method preserves the traditional fibrous texture while ensuring microbiological safety and a prolonged shelf life.

Added Section

Supplementary Information

S1. Detailed Production Procedure of Chechil Cheese

Traditional Production (Erzurum method)

Raw cow’s milk (average fat content: 3.6%) was filtered through a stainless-steel mesh and heated to 30 ± 1 °C in a double-jacket vat. Calf rennet paste (strength: 1:15,000) was diluted 1:10 with distilled water and added at a ratio of 2 mL per 10 L of milk. Coagulation was allowed to proceed for 60 ± 5 min until a firm curd was formed. The curd was cut into cubes (~1 cm³) using stainless steel knives and rested for 5 min. Whey drainage was performed by gentle stirring for 10 min, followed by filtration through cheesecloth for 15 min. The curd was scalded in hot whey at 75 ± 2 °C for 15 min until it became elastic, then manually stretched and braided into strands approximately 40–50 cm in length. The cheese was dry-salted at 2.5% (w/w) and air-dried at 10 ± 2 °C and 75% RH for 48 h.

Industrial Production

Standardized cow’s milk (fat content adjusted to 3.2%) was pasteurized at 65 °C for 30 min and cooled to 30 ± 1 °C. A commercial mesophilic starter culture (0.02%, w/v) and microbial rennet (strength: 1:18,000) diluted 1:10 were added at 2 mL per 10 L milk. Coagulation was achieved within 45 ± 3 min. The curd was cut into 0.5 cm cubes and stirred gently for 20 min while heating to 38 °C. Whey was drained through a perforated stainless-steel plate, and curd blocks were scalded at 80 °C for 10 min in brine whey (3% NaCl). Fibrous structure was developed by mechanical stretching, followed by shaping into ~500 g portions. The cheese was brined at 8% NaCl for 24 h at 8 °C, then dried at 12 ± 1 °C and 75% RH for 36 h before vacuum packaging.

Line 158. The number of table “Table 17” is not correct.

Response : In accordance with the reviewer’s comment, all table numbers have been reviewed and renumbered sequentially, starting from Table 1, according to their order of citation in the text.

The abstract is extremely long; please review the journal’s guidelines

Response: The abstract of the manuscript has been revised as specified below, resulting in a total of 197 words.

Extending the shelf life and ensuring microbial stability of processed foods are key objectives in the food industry. In this study, edible films containing chitosan, chitosan + thyme oil, and chitosan + rosemary oil were applied to traditional and industrial Cecil cheeses using the dipping method, with control groups for each production type. Samples were stored at 4 ± 1 °C for 45 days, and physical (color, water activity, texture), chemical (pH, acidity, dry matter), microbiological (total aerobic mesophilic bacteria, yeast-mold, coliforms, lactic acid bacteria), and sensory analyses were performed on days 1, 15, 30, and 45. Results indicated that chitosan-based films effectively limited microbial growth, with the chitosan + rosemary oil combination being particularly effective in reducing microbial load and maintaining textural stability. Traditional cheeses achieved higher overall acceptability, while purchase intent was greater for industrial products. Coated samples exhibited slower pH decline and more stable dry matter content, with industrial cheeses retaining moisture more effectively. Texture profile analysis showed more stable chewiness and springiness values in coated samples. In conclusion, natural edible films represent an effective approach for extending shelf life and preserving quality, particularly in traditional cheeses with fibrous structures and shorter shelf life.

Please verify the maximum number of keywords permitted.

Response: In accordance with the reviewer’s suggestion, the keywords have been updated as follows: Keywords: Chitosan, Essential oils, Cecil cheese, Microbiological quality, Sensory evaluation, Bioactive coating.

Please specify the degree of deacetylation of the chitosan.

Response: We thank the reviewer for this valuable comment. The chitosan used in this study has a degree of deacetylation of 85%, and this information has been added to the Materials and Methods section of the manuscript.

Added Section (2.1. Materials)

For the preparation of edible films, chia seeds were purchased in bulk from a local herbal store, while chitosan (degree of deacetylation of 85 %), was obtained from TİENS İç ve Dış Tic. Ltd. Şti. Thyme and rosemary essential oils used in film formulations were acquired from the brand KIRINTI (Kırıntı Baharat Hay. Tar. Kozm. Gıda Tem. Mlz. İnş. Elek. Tur. San. Tic. Ltd. Şti.).

Change "Chia Seed and Rosemary Oil Combination" to "chia seed and rosemary oil combination" (all lowercase).

Response: We thank the reviewer for this comment. The phrase “Chia Seed and Rosemary Oil Combination” has been changed to “chia seed and rosemary oil combination” (all lowercase) throughout the manuscript.

Review the manuscript carefully to apply this rule uniformly for all similar cases.

Response: We thank the reviewer for this valuable suggestion. The manuscript has been carefully reviewed, and the same formatting rule has been applied consistently to all similar cases throughout the text.

Please specify how the essential oils were obtained.

Response: We thank the reviewer for this valuable comment. The essential oils used in this study were commercially obtained from Kırıntı Baharat Hayvansal-Tarımsal-Kozmetik-Gıda-Temizlik Mlz. Elektrik Turizm San. Tic. Ltd. Şti., and this information has been added to the Materials section of the manuscript.

Added Section (Material)

Thyme and rosemary essential oils used in film formulations were acquired from the brand KIRINTI (Kırıntı Baharat Hay. Tar. Kozm. Gıda Tem. Mlz. İnş. Elek. Tur. San. Tic. Ltd. Şti.).

Please ensure that ‘Cecil’ is spelled consistently throughout the manuscript.

Response: We thank the reviewer for this valuable comment. The spelling of “Cecil” has been carefully checked and corrected to ensure consistency throughout the entire manuscript. The parts previously written as “Çeçil” have been changed to “Cecil.”

Align Tables 3 and 4.

Response: We thank the reviewer for this comment. All tables have been carefully checked and aligned consistently throughout the manuscript.

Check Table 5 for spacing and information.

Response: We thank the reviewer for this comment. All tables have been carefully reviewed, and the requested adjustments regarding spacing and information have been implemented.

Tables 5 and 6 and 7, express the average number of measurements.

Response: We thank the reviewer for this valuable comment. The requested tables have been updated to present the data as mean ± standard deviation.

Arrange the figures so that each fits entirely on a single page.

Response: We thank the reviewer for this comment. All figures have been adjusted to fit entirely on a single page, and the changes have been applied both in the manuscript and in the Figure section.

Added Section (Figures)

Figures

Fig.1 Traditional Çeçil cheese production flowchart

Fig. 2 Industrial Production Flow Chart

However, there is too much content for a single paper. It might be better to focus on either the preservation of industrial-type cheeses or traditional-type cheeses, but not both.

Response: We sincerely thank the reviewer for the valuable comments. In this study, we aimed to comprehensively investigate the effects of both traditional and industrial production methods, as well as different coatings, on the physicochemical, microbiological, sensory, and textural properties of Chechil cheese. Through this approach, we intended to fill a gap in the literature and provide data that could guide future research. Although we acknowledge that the study scope is extensive, it has allowed us to achieve our objectives and evaluate the topic in a holistic manner. In accordance with your suggestion, we will consider focusing on a single production type in future studies.

Remove figures 5 to 10 and label the equipment in the appropriate places.

Response: In accordance with the reviewer’s suggestion, Figures 6–10 have been removed from both the manuscript and the Figures section. Additionally, the equipment has been appropriately labeled in the relevant places.

It is suggested to include photos showing the preservation over time.

Response: In accordance with the reviewer’s suggestion, photos showing the preservation of the cheese over time have been added to the manuscript.

Added Section (Figures)

Fig. 6 Traditional and industrially produced Cecil cheese during storage days

Express the essential oils using their scientific names the first time they are mentioned.

Response: In accordance with the reviewer’s suggestion, the essential oils have been expressed with their scientific names the first time they are mentioned in the manuscript. In the Abstract section, thyme oil (Thymus vulgaris) and rosemary oil (Rosmarinus officinalis) are presented with their scientific names at their first mention.

Added Section (Abstract)

Extending the shelf life and ensuring microbial stability of processed foods are key objectives in the food industry. In this study, edible films containing chitosan, chitosan + thyme (Thymus vulgaris) oil, and chitosan + rosemary (Rosmarinus officinalis) oil were applied to traditional and industrial Cecil cheeses using the dipping method, with control groups for each production type.

Indicate the ATCC or identification of the strains.

Response: In accordance with the reviewer’s suggestion, the ATCC numbers or identifications of the microbial strains used in the study have been added to the manuscript.

Added Section (Material)

The starter cultures used in cheese production were obtained in powdered form as commercial preparations of the “Danisco” brand, supplied by Türker Endüstri Teknik Makine ve Tic. Ltd. Şti. The cultures were prepared in accordance with the procedures indicated on the manufacturer’s packaging and were applied as mixed cultures during cheese production. The culture compositions employed in the study were as follows:

Culture 1: Lactococcus lactis subsp. lactis ( ATCC 19435) + Lactococcus lactis subsp. cremoris ( ATCC 19257 ) + Lactococcus lactis subsp. diacetylactis (ATCC 13675), Culture 2: Lactococcus lactis subsp. lactis ( ATCC 19435) + Lactococcus lactis subsp. cremoris ( ATCC 19257 ), Culture 3: Streptococcus salivarius subsp. thermophilus ( ATCC 19258).

Chechil cheese sometimes appears as. Correct it.

Response: In accordance with the reviewer’s suggestion, the phrase “Chechil cheese sometimes appears as” has been corrected in the manuscript.

Line 493. Figure 10 does not represent the technique described here at all.

Response: In accordance with the reviewer’s suggestion, all figures from Figure 6 to Figure 10 have been removed from both the manuscript and the Figures section.

Section 3.2 is repeated in all the subsequent items. It would be advisable to separate them as 3.2.1, 3.2.2, and so on.

Response: In accordance with the reviewer’s suggestion, the subsections of Section 3.2 have been renumbered as 3.2.1, 3.2.2, and so on.

Review the English throughout the entire manuscript.

Response: In accordance with the reviewer’s suggestion, the English language throughout the manuscript has been carefully reviewed and necessary corrections have been made.

Round 2

Reviewer 1 Report

Comments and Suggestions for Authors

Accept

Reviewer 2 Report

Comments and Suggestions for Authors

The paper can be accepted, as the quality has been significantly improved.

Reviewer 3 Report

Comments and Suggestions for Authors

The requested corrections have been made. I suggest approving the work.